# Variation in bradyrhizobial NopP effector determines symbiotic incompatibility with *Rj2*-soybeans via effector-triggered immunity

Masayuki Sugawara[1], Satoko Takahashi[1], Yosuke Umehara[2], Hiroya Iwano[1], Hirohito Tsurumaru[3], Haruka Odake[1], Yuta Suzuki[1], Hitoshi Kondo[1], Yuki Konno[1], Takeo Yamakawa[4], Shusei Sato[1], Hisayuki Mitsui[1] & Kiwamu Minamisawa[1]

Genotype-specific incompatibility in legume–rhizobium symbiosis has been suggested to be controlled by effector-triggered immunity underlying pathogenic host-bacteria interactions. However, the rhizobial determinant interacting with the host resistance protein (e.g., Rj2) and the molecular mechanism of symbiotic incompatibility remain unclear. Using natural mutants of *Bradyrhizobium diazoefficiens* USDA 122, we identified a type III-secretory protein NopP as the determinant of symbiotic incompatibility with *Rj2*-soybean. The analysis of *nopP* mutations and variants in a culture collection reveal that three amino acid residues (R60, R67, and H173) in NopP are required for *Rj2*-mediated incompatibility. Complementation of *rj2*-soybean by the *Rj2* allele confers the incompatibility induced by USDA 122-type NopP. In response to incompatible strains, *Rj2*-soybean plants activate defense marker gene *PR-2* and suppress infection thread number at 2 days after inoculation. These results suggest that *Rj2*-soybeans monitor the specific variants of NopP and reject bradyrhizobial infection via effector-triggered immunity mediated by Rj2 protein.

[1] Graduate School of Life Sciences, Tohoku University, 2-1-1 Katahira, Aoba-ku, Sendai, Miyagi 980-8577, Japan. [2] Institute of Agrobiological Sciences, National Agriculture and Food Research Organization, 2-1-2 Kannondai, Tsukuba, Ibaraki 305-8602, Japan. [3] Faculty of Agriculture, Kagoshima University, 1-21-24 Korimoto, Kagoshima 890-0065, Japan. [4] Faculty of Agriculture, Graduate School of Bioresource and Bioenvironmental Sciences, Kyushu University, 6-10-1 Hakozaki, Higashi-ku, Fukuoka 812-8581, Japan. Correspondence and requests for materials should be addressed to M.S. (email: msugawara@ige.tohoku.ac.jp)

The type III secretion system (T3SS) is common to bacterial pathogens of plants and animals[1,2]. It delivers effector proteins directly into the host cells[1,2] and helps the pathogen to survive and to escape the immune response activated by microbe/pathogen-associated molecular patterns. To counter the virulence functions of effectors, plants have evolved resistance (R) genes encoding nucleotide-binding site/leucine-rich repeat (NBS-LRR) proteins that often directly or indirectly recognize specific effectors and mount a strong immune response called effector-triggered immunity (ETI)[3].

Rhizobia symbiotically reduce atmospheric dinitrogen in root nodules to ammonia that is utilized by the host legumes, resulting in improved plant growth for sustainable agriculture[4,5]. During the establishment of symbiosis, rhizobia generally produce lipochitin oligosaccharides, called Nod factors (NFs), in response to plant flavonoid compounds to activate host signaling, leading to rhizobial infection and nodule organogenesis[6]. Symbiotic specificity between legumes and rhizobia is initially determined by the structure of NFs and host NF receptor kinases (e.g., NFR1 and NFR5 in *Lotus japonicus*)[7,8]. However, many rhizobia have genes encoding T3SS structural components (*rhc* genes) such as pathogens and the effector proteins often stimulate the host immune system and alter the host range[9–12].

A number of rhizobial T3SS effector proteins have been identified as nodulation outer proteins (Nops)[10,13]. Flavonoids derived from host plants activate a transcriptional regulator NodD, which induces the expression of *ttsI*, a gene for a positive transcriptional regulator of the *rhc* and *nop* genes[14,15]. Several Nops suppress plant defense reactions and hence promote nitrogen-fixing nodule symbiosis[16]. By genomic analysis, more than 30 genes in *Bradyrhizobium* and approximately 15 genes in *Sinorhizobium fredii* have been predicted to encode T3SS effectors[17], but their symbiotic and biochemical functions have not been fully elucidated.

Soybean [*Glycine max* (L.) Merr.] is a major leguminous crop worldwide. The plants form root nodules upon infection with nitrogen-fixing bradyrhizobia, such as *Bradyrhizobium diazoefficiens*, *Bradyrhizobium japonicum*, and *Bradyrhizobium elkanii*[18,19]. Several dominant genes (*Rj2*, *Rj3*, *Rj4*, and *Rfg1*) in soybeans restrict nodulation with specific rhizobial strains[20]. Among them, *Rj2*-genotype soybeans restrict nodulation with specific *Bradyrhizobium* strains, including *B. diazoefficiens* USDA 122[21,22]. Positional cloning has revealed that the *Rj2* gene encodes a typical R protein of the Toll-interleukin receptor/NBS/LRR (TIR-NBS-LRR) class[23]. This finding indicates that symbiotic incompatibility between USDA 122 and *Rj2*-soybean plants may be controlled by ETI. The inactivation of the T3SS in *B. diazoefficiens* USDA 122 restores its nodulation capability on the *Rj2*-soybean cultivar Hardee[24]. The deduced amino acid sequences of proteins encoded by the *rhc* and *ttsI* genes are identical between *B. diazoefficiens* USDA 122 and the compatible strain *B. diazoefficiens* USDA 110[T], suggesting that a T3SS-dependent effector protein in USDA 122 block nodulation with *Rj2*-soybean plants[24]. However, the T3SS effector and the molecular mechanism of this genotype-specific incompatibility have been unknown.

Here we have obtained spontaneous mutants of *B. diazoefficiens* USDA 122 that overcome *Rj2*-mediated incompatibility and identified NopP as a causal effector of this incompatibility. We also reveal that, among different amino acid residues of NopP in strain USDA 122 and a compatible strain USDA 110[T], three amino acid residues (R60, R67, and H173) are required for inducing this symbiotic incompatibility via ETI mediated by Rj2 protein.

## Results

**Nodulation phenotypes of USDA 122 and its T3SS mutants.** The *Rj2*-genotype soybean cultivar Hardee restricts nodulation with wild-type *B. diazoefficiens* USDA 122 but not with USDA 122 carrying mutated genes for a T3SS structural component (122Ω*rhcJ*) and a positive transcriptional regulator of T3SS-related genes (122Ω*ttsI*)[24]. Consistent with this report, Hardee plants inoculated with 122Ω*rhcJ* and 122Ω*ttsI* formed significantly more nodules than those inoculated with wild-type USDA 122; the number of nodules induced by USDA 110[T] was comparable to that of 122Ω*rhcJ* and 122Ω*ttsI* (Fig. 1a). As the plants inoculated with 122Ω*rhcJ* or 122Ω*ttsI* grew well and had green leaves in nitrogen-free medium (Fig. 1b), the corresponding nodules apparently fixed N₂.

Wild-type USDA 122 formed no or only a single nodule at a low frequency and its nodulation phenotype was clearly distinct from those of 122Ω*rhcJ* and 122Ω*ttsI* (Fig. 1a, c, d). We considered two explanations for the sporadic nodulation on Hardee roots: (i) a few cells of wild-type USDA 122 passed through the barrier of *Rj2*-mediated incompatibility or (ii) the single nodules were formed by spontaneous USDA 122 mutants with defects in genes responsible for *Rj2*-mediated incompatibility.

**Spontaneous mutants overcoming *Rj2*-mediated incompatibility.** We inoculated 26 Hardee plants with wild-type USDA 122, purified eight isolates (W1-1a, W3-1a, W3-2a, W8-1a, W8-1b, W9-1a, W9-1b, and W9-3a) from eight independent nodules, and re-inoculated the isolates onto Hardee. All isolates consistently formed more nodules on Hardee roots (Nod[+] phenotype) than did USDA 122 (Fig. 2a, b), clearly showing that they were all spontaneous mutants of USDA 122 with defects in genes necessary to trigger *Rj2*-mediated incompatibility. However, nitrogen fixation phenotypes differed among the eight isolates, as indicated by acetylene reduction assay (Fig. 2a) and leaf color (Fig. 2b). Strains W3-1a, W9-1a, and W9-1b elicited normal nitrogen-fixing nodules (Nod[+]/Fix[+] phenotype), whereas W1-1a, W3-2a, W8-1a, W8-1b, and W9-3a showed no nitrogen fixation activity (Nod[+]/Fix[−] phenotype).

To search for the mutations, we mapped MiSeq reads of the eight mutants on the complete genome sequence of USDA 122[25]. In the genomes of the three Nod[+]/Fix[+] mutants, we identified several single-nucleotide polymorphisms (SNPs) and insertions of the insertion sequence (IS) elements ISRj1, ISRj2, and ISBdi2 (Supplementary Table 1). W3-1a and W9-1a had mutations in the *nopP* gene within the *nif* gene cluster in a symbiosis island, caused by different IS element insertions (Fig. 2c, Supplementary Fig. 1). As *nopP* encodes a T3SS effector[26,27], we considered it a candidate gene for *Rj2* symbiotic incompatibility. On the other hand, W9-1b had a SNP within the BD122_09230 locus in a gene cluster including the gene encoding the T3SS structural component *rhcV* (Fig. 2c), suggesting that this mutant might lack a functional T3SS.

In the genomes of the five Nod[+]/Fix[−] strains, we found ISRj1- and ISRj2-mediated deletions that included both *nif* and *rhc* gene clusters (Fig. 2c, Supplementary Figs. 2–5). As the *nif* genes are required for nitrogen fixation, these mutants showed the Fix[−] phenotype in addition to the Nod[+] phenotype caused by the lack of the *rhc* genes, which are required for T3SS function.

**Functional validation of *nopP* on symbiotic incompatibility.** *nopP* is located in the center of the *nif* gene cluster (Fig. 3a), and a *cis* element motif (*tts*-box) is present upstream (−100 to −63) of *nopP* (Fig. 3b). Genistein, a flavonoid inducer of T3SS genes in bradyrhizobia, greatly increased NopP protein secretion via T3SS

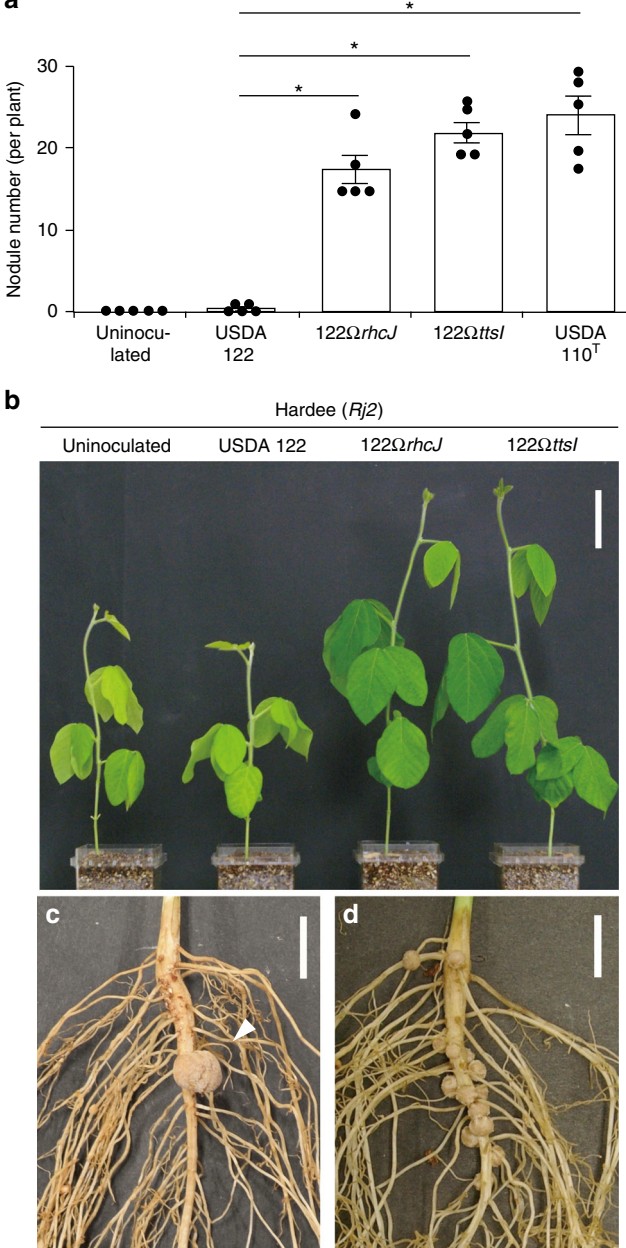

**Fig. 1** Symbiotic phenotypes of Hardee (*Rj2*) inoculated with *B. diazoefficiens* strains. **a** Number of nodules formed on roots at 28 days after rhizobial inoculation. No experimental repeat was conducted with the same comparison. Error bars show SEM (*n* = 5). *Significant difference from USDA 122 (*P* < 0.01 by Dunnett's multiple comparison test). **b** Shoots 28 days after inoculation with the indicated strains. Scale bar, 5 cm. **c** Roots of plants inoculated with USDA 122. The arrow indicates a single sporadic nodule; normal nodules were rarely formed by USDA 122. Scale bar, 1 cm. **d** Roots of plants inoculated with 122Ω*rhcJ*. Scale bar, 1 cm

in USDA 122 but its secretion was not detected in 122Ω*rhcJ* (Fig. 3c). These results indicate that *nopP* encodes a T3SS effector protein.

To examine whether NopP induces symbiotic incompatibility with *Rj2*-soybeans, we constructed an in-frame *nopP* deletion mutant 122Δ*nopP* (Fig. 3a) and inoculated it onto Hardee. Plants inoculated with 122Δ*nopP* formed significantly more nodules than plants inoculated with wild-type USDA 122 (Fig. 3d, e). The symbiotic phenotype of 122Δ*nopP* was similar to that of a T3SS

mutant (122Ω*rhcJ*) in terms of nodulation and nitrogen fixation (Fig. 3d–f). The efficient symbiotic phenotype with 122Δ*nopP* was also observed in other *Rj2*-soybean cultivars, CNS and IAC-2 (Supplementary Fig. 6a−d). In an *rj2*-soybean cultivar Lee, the symbiotic phenotypes were not different between inoculations of USDA 122 and 122Δ*nopP* (Supplementary Fig. 6e, f). These results clearly indicate that NopP effector in *B. diazoefficiens* USDA 122 is necessary to trigger *Rj2* symbiotic incompatibility.

**nopP allele swapping between USDA 122 and USDA 110ᵀ.** Although NopP of USDA 122 is a T3SS effector inducing *Rj2* symbiotic incompatibility, the *nopP* gene is also present in *B. diazoefficiens* USDA 110ᵀ (Supplementary Fig. 7a) but does not induce *Rj2*-mediated incompatibility. Moreover, the NopP protein was secreted from USDA 110ᵀ cells when genistein is added to culture medium (Fig. 4a) via T3SS[27], suggesting that symbiotic compatibility between bradyrhizobia and *Rj2*-soybeans is not determined by secretion of NopP but by its structure.

An alignment of NopP amino acid sequences (277 aa) showed substitutions of four amino acid residues between USDA 122 and 110ᵀ (Fig. 4b, Supplementary Fig. 7b). To examine whether the difference determines symbiotic compatibility, we swapped the *nopP* genes between USDA 122 and 110ᵀ. A USDA 122 derivative carrying USDA 110-type *nopP* (122*nopP*₁₁₀) formed significantly more nodules on Hardee than did wild-type USDA 122 and the leaves of 122*nopP*₁₁₀ were green, indicating nitrogen-fixing activity of the nodules (Fig. 4c, d). A USDA 110ᵀ derivative carrying USDA 122-type *nopP* (110*nopP*₁₂₂) showed a nodulation-deficient phenotype similar to that of wild-type USDA 122 (Fig. 4c, d). 110*nopP*₁₂₂ secreted the NopP protein as well as the wild-type USDA 110ᵀ (Fig. 4a), suggesting that substitutions of NopP on the amino acid residues at positions 60, 67, 173, and 271 does not affect the ability of the bacteria to deliver the protein. These results indicate that the difference of four amino acid residues in NopP between USDA 122 and 110ᵀ determines *Rj2* symbiotic incompatibility.

To examine when *Rj2*-soybeans block the infection of USDA 122, we microscopically observed early infection events (Supplementary Fig. 8). The number of infection thread (IT) formed on the main roots inoculated with incompatible strains (USDA 122 or 110*nopP*₁₂₂) was significantly less than those with compatible strains (122*nopP*₁₁₀ or USDA 110ᵀ) at 2 days after inoculation (DAI) (Fig. 4e). These results suggest that *Rj2*-soybeans mainly block the incompatible rhizobia at a step before IT formation.

**NopP-mediated symbiotic incompatibility through Rj2 protein.** The host Rj2 protein is a TIR-NBS-LRR class of R protein, and its E452 and I490 residues located between the NBS and LRR domains are reportedly required for incompatibility with USDA 122[23]. To investigate the involvement of Rj2 in NopP-mediated symbiotic incompatibility, we produced transgenic soybean cultivar Lee (*rj2*) plants complemented with the complementary DNA of *Rj2* (Hardee) or *rj2* (Hardee, E452K, I490R) and evaluated their nodulation phenotypes by inoculating them with USDA 122 and 122*nopP*₁₁₀. As hairy root transformation was conducted without antibiotic selection, the resulting hairy roots were either transgenic or wild type, which could be readily distinguished by examining the presence of green fluorescent protein (GFP) encoded in the used binary vector[28].

When *Rj2*-transformed roots were inoculated with USDA 122, nodules were rarely observed on transgenic hairy roots (Fig. 5a, b). In contrast, when they were inoculated with 122*nopP*₁₁₀, nodule formation was observed (Fig. 5c, d). Roots transformed with *rj2* or empty vector formed nodules regardless of whether it was infected with USDA 122 or 122*nopP*₁₁₀ (Fig. 5e

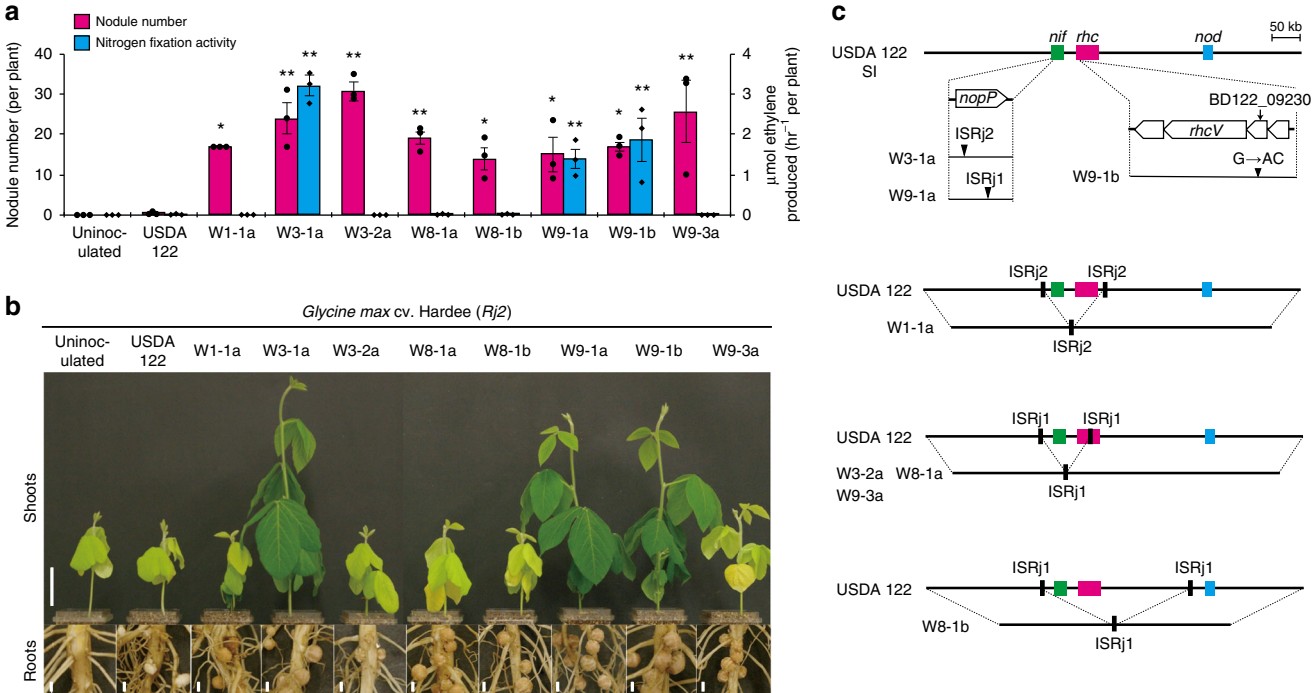

**Fig. 2** Characterization of spontaneous mutants overcoming *Rj2*-mediated incompatibility. **a** Number of root nodules and nitrogen fixation activity of the roots of *G. max* cv. Hardee at 28 days after inoculation with USDA 122 and its spontaneous mutants. The data of nodule number shown are representative of two independent experiments. Nitrogen fixation activity was measured by acetylene reduction assay. Error bars show SEM ($n = 3$). Significant differences from USDA 122 by Dunnett's multiple comparison test: *$P < 0.05$, **$P < 0.01$. **b** Shoots and roots of *G. max* cv. Hardee (*Rj2*) at 28 days after inoculation with USDA 122 or its spontaneous mutants. Scale bars, 5 cm (for shoots) or 0.5 cm (for roots). **c** Positions of mutations in the genomes of spontaneous mutants. Bold horizontal lines indicate the symbiosis island (SI, 672 kb). ISRj1 and ISRj2, insertion sequence elements

−l). The average number of nodules was significantly lower on *Rj2* transgenic roots inoculated with USDA 122 than on those inoculated with $122nopP_{110}$, whereas the average nodule number on roots transformed with *rj2* or empty vector was similar after inoculation with USDA 122 and $122nopP_{110}$ (Fig. 5m). These results indicate that the residues at positions 452 and 490 of Rj2/ rj2 determine the symbiotic compatibility with *B. diazoefficiens* strains having 122-type NopP. Therefore, NopP-mediated *Rj2* symbiotic incompatibility is induced via the host Rj2 protein.

**Expressions of defense marker genes in *Rj2*-soybean roots**. To examine whether defense responses are activated under *Rj2* symbiotic incompatibility, we monitored the expressions of defense marker genes (*PR-1*, *PR-2*, and *PR-5*) in Hardee roots. The expression of *PR-2* was significantly increased at 2 DAI with USDA 122 as compared with $122nopP_{110}$ inoculation and uninoculated control (Fig. 6), and markedly decreased at 4 DAI, suggesting the transient and early expression of *PR-2* elicited by wild-type USDA 122 (Fig. 6). In contrast, the expression of *PR-1* and *PR-5* was not significantly changed (Supplementary Fig. 9). A significant change in *Rj2* expression was not observed (Fig. 6) as reported previously[23]. The expression of *ENOD40*, a well-characterized symbiosis gene that is expressed after the NF perception[29], was induced in Hardee roots at 4 DAI by the inoculation with both USDA 122 and $122nopP_{110}$ strains (Fig. 6). These results suggested that, in *Rj2* symbiotic incompatibility, NF perception and further symbiosis signaling is activated but failed to induce normal nodulation by host defense response.

**Amino acid residues of NopP responsible for incompatibility**. To determine which of the four amino acid residues in NopP that differed between USDA 122 and 110[T] (Fig. 4b) are responsible for

*Rj2* symbiotic incompatibility, we complemented the *nopP* deletion mutant of USDA 110[T] (110Δ*nopP*) with DNAs encoding each of the 16 possible NopP variants (110*nopP*_v1 to 110*nopP*_v16) that differed in these four residues.

The number of nodules formed on Hardee (*Rj2*) roots after inoculation with the 16 strains is shown in Fig. 7. Plants inoculated with USDA 110[T] and 110Δ*nopP* formed 23–25 nodules per plant, whereas plants inoculated with 110*nopP*_v1, which had USDA 122-type *nopP*, formed very few nodules, similar to those inoculated with USDA 122 or $110nopP_{122}$ (Fig. 4). Plants inoculated with 110*nopP*_v1 to 110*nopP*_v4, which have R60 and H173, formed significantly fewer nodules than those inoculated with 110Δ*nopP*. Slightly fewer nodules were formed upon inoculation with 110*nopP*_v1 or 110*nopP*_v2 than with 110*nopP*_v3 or 110*nopP*_v4. Indeed, the second experiments with 10 replicates showed that Hardee inoculated with 110*nopP*_v2 formed significantly fewer nodules than those inoculated with 110*nopP*_v4 ($P = 0.022$ by two-tailed Student's *t*-test) (Supplementary Fig. 10a). Nodule number was not affected by the amino acid residue at position 271 (F or L) or by L60 (110*nopP*_v9 to 110*nopP*_v16). Plants inoculated a derivative of USDA 122 carrying 110*nopP*_v4-type *nopP* (122*nopP*_m3) showed apparently intermediate nodule numbers as compared with $122nopP_{110}$ and USDA 122 (Supplementary Fig. 10b). Taken together, these results indicate that having R60, R67, and H173 in NopP are required for the suppression of root nodule formation on *Rj2*-soybeans.

**Diversity and phylogeny of NopP among soybean bradyrhizobia**. To evaluate *nopP* diversity, we searched the NCBI GenBank database for *nopP* homologs from *B. diazoefficiens* and *B. japonicum* (Supplementary Table 2). In

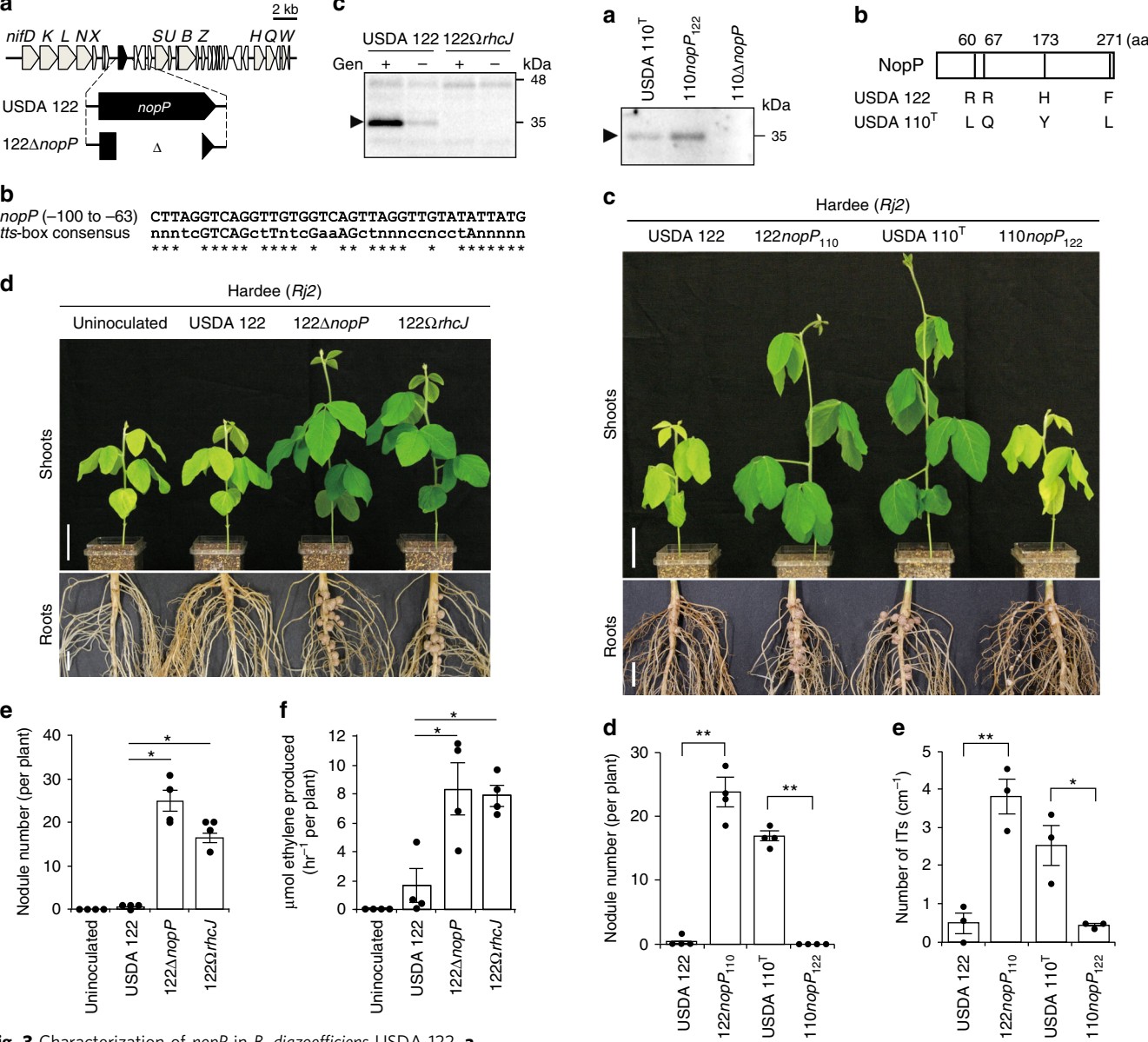

**Fig. 3** Characterization of *nopP* in *B. diazoefficiens* USDA 122. **a** Organization of *nopP* and its flanking regions in USDA 122 and 122Δ*nopP*. **b** The *tts*-box motif found upstream of *nopP* in USDA 122 and the *tts* consensus sequence[64]. Asterisks indicate identical nucleotides. **c** Western blotting analysis of NopP secreted into culture supernatants from USDA 122 and its mutant with inactivated T3SS (122Ω*rhcJ*). Strains were grown in the presence (+) or absence (–) of 10 μM genistein (Gen). The position of NopP is indicated by arrow. **d** Shoots and roots of *G. max* cv. Hardee at 28 days after inoculation with USDA 122, 122Δ*nopP*, or 122Ω*rhcJ*. Scale bars, 5 cm (for shoots) or 1 cm (for roots). **e** Number of root nodules and **f** nitrogen fixation activity. The data of nodule number shown are representative of three independent experiments. Error bars show SEM ($n = 4$). *Significant difference from USDA 122 ($P < 0.01$ by Dunnett's multiple comparison test)

**Fig. 4** Effect of *nopP* swapping between USDA 122 and 110[T] on symbiosis with Hardee. **a** Western blotting analysis of NopP secreted into culture supernatants from USDA 110, 110*nopP*₁₂₂, and *nopP* deletion mutant of USDA 110[T] (110Δ*nopP*). Strains were grown in the presence of 10 μM genistein. The position of NopP is indicated by arrow. **b** Differences in amino acid residues in NopP between *B. diazoefficiens* USDA 122 and 110[T]. **c** Shoots and roots of cv. Hardee at 28 days after inoculation with USDA 122, 110[T], or strains with swapped *nopP* (122*nopP*₁₁₀, 110*nopP*₁₂₂). Scale bars, 5 cm (for shoots) or 1 cm (for roots). **d** Number of nodules formed on roots at 28 days after inoculation. The data shown are representative of three independent experiments. Error bars show SEM ($n = 4$). **e** Number of infection threads (ITs). ITs were counted on whole region of main roots at 2 days after inoculation. Error bars show SEM ($n = 3$). *$P < 0.05$, **$P < 0.01$ by two-tailed Student's *t*-test

parallel, we selected phylogenetically diverse *B. diazoefficiens* and *B. japonicum* strains from our culture collection[30] (Supplementary Table 2) and sequenced their *nopP* genes. The deduced amino acid sequences from a total of 55 strains were aligned and classified (Fig. 8a, Supplementary Table 3). The NopP sequence pattern fell into nine types, including NopP of USDA 110[T] and 122. The six newly identified types, designated ST1 to ST6, had the same length (277 aa) as the 110 and 122

types, and one type was disrupted by the insertion of an IS element (designated "IS-inserted"). Besides positions 60, 67, 173, and 271, 14 other variations were observed from ST3 to ST6 types (Supplementary Table 3).

In addition to *B. diazoefficiens* strains, some *B. japonicum* strains also have the 122-type gene (Supplementary Table 3).

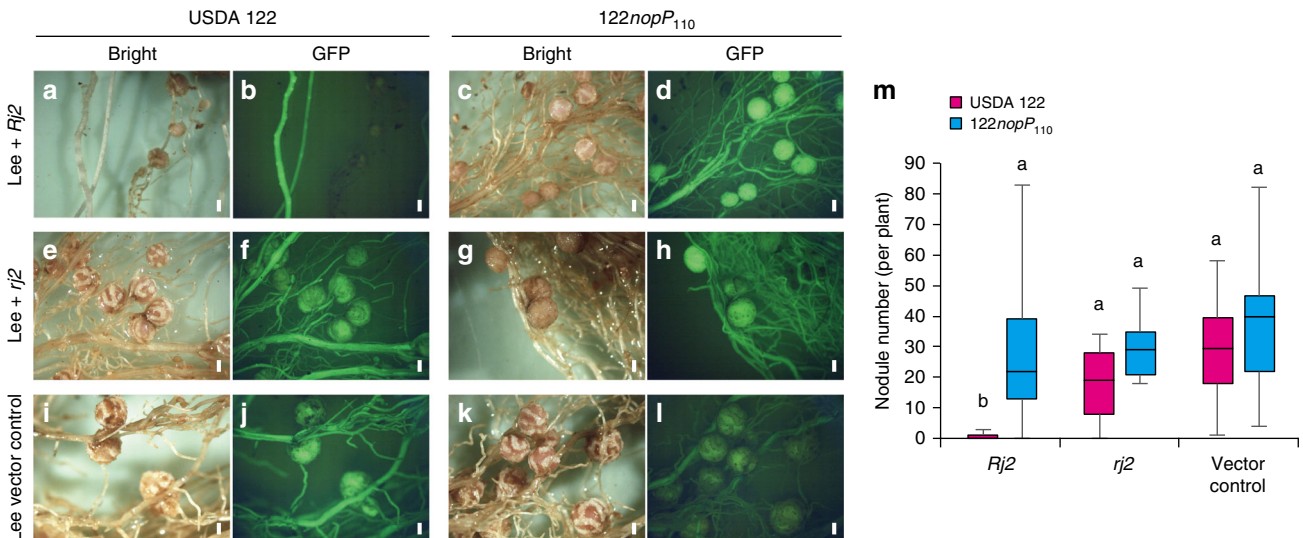

**Fig. 5** Nodulation of *rj2*-soybean transformed with the *Rj2* gene by USDA 122 or 122*nopP*₁₁₀. **a**, **c**, **e**, **g**, **i**, **k** Bright-field images and **b**, **d**, **f**, **h**, **j**, **l** GFP fluorescence images of roots of transgenic cv. Lee (*rj2*) complemented with the *Rj2*, *rj2* gene or vector control. *B. diazoefficiens* USDA 122 or 122*nopP*₁₁₀ was inoculated onto each seedling, and images were taken at 4 weeks after inoculation. Scale bars, 1 mm. **m** Box-and-whisker plots showing the number of nodules formed on GFP-expressing hairy roots. Combined data from three independent experiments are shown. Center line, median; box limits, first and third quartiles; whiskers, ranges. *n* = 14 for *Rj2*, *n* = 12 for *rj2*, and *n* = 13 for vector control. Different letters above bars indicate statistical significance (*P* < 0.01 by nonparametric Steel–Dwass multiple comparison test)

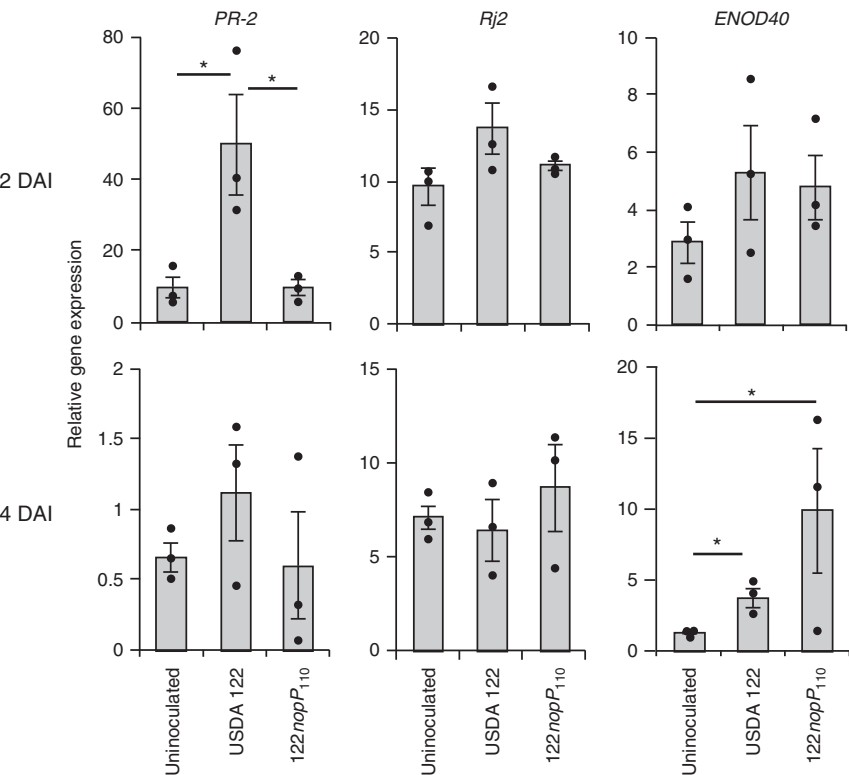

**Fig. 6** Expression of *PR-2*, *Rj2*, and *ENOD40* genes in roots of Hardee. Expression of *PR-2*, *Rj2*, and *ENOD40* in roots of *G. max* cv. Hardee inoculated with USDA 122 and 122*nopP*₁₁₀ was determined by quantitative reverse-transcription PCR using primers listed in Supplementary Data 2 as described in Methods. RNA was isolated from the roots at 2 and 4 days after inoculation (DAI). The expression level of each gene was normalized by the *SUBI2* (ubiquitin) gene. Error bars show SEM of three independent experiments with two plants (from one seed pack). *P* < 0.05 by two-tailed Student's *t*-test

Such *B. japonicum* strains (HK7-6, TS4-11, and J5), as well as the ST1-type *B. japonicum* strain KW1-63 (corresponding to 110*nopP*_v2 in Fig. 7), did not nodulate on Hardee (Supplementary Fig. 11, Fig. 8a). The NopP sequences of these strains contained R60, R67, and H173 residues, and their symbiotic phenotypes were consistent with the results shown in Fig. 7. Together with the analysis of *nopP* mutational variants (Fig. 7), these results indicate that a combination of

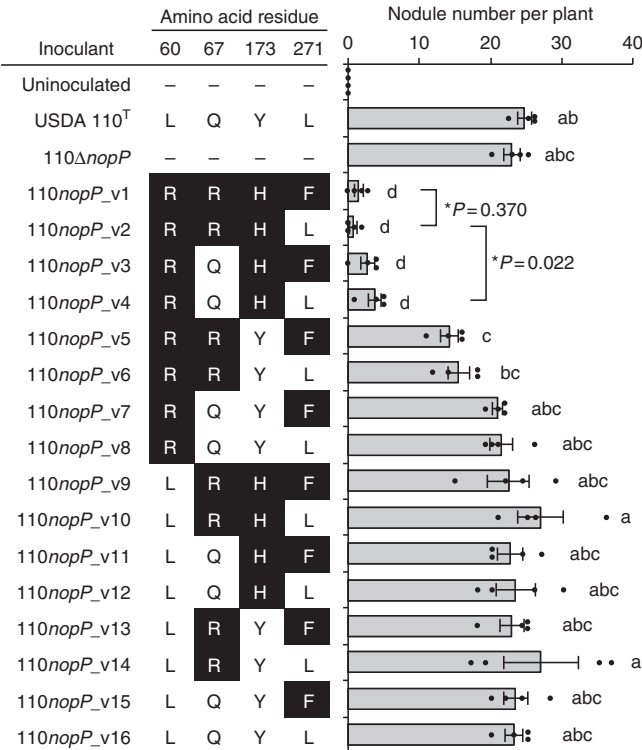

**Fig. 7** Nodulation phenotypes of USDA 110[T] carrying *nopP* variants on Hardee. Strains and amino acid residues in NopP at positions 60, 67, 173, and 271 are shown on the left, and the nodule numbers at 28 days after inoculation are shown on the right. Error bars show SEM ($n = 4$). Different lowercase letters to the right of the bars indicate significant difference by one-way ANOVA followed by Tukey's honestly significant difference test ($P < 0.05$). The experiment was repeated once again with different comparison as second nodulation assay (Supplementaly Fig. 10a). The asterisks indicate the *P*-values calculated by two-tailed Student's *t*-test based on the data from second nodulation assay ($n = 10$)

R60, R67, and H173 is critical for the induction of *Rj2* symbiotic incompatibility.

Next, we mapped the NopP types (122, 110, ST1, and ST2) on a phylogenetic tree of bradyrhizobia based on the 16S–23S rRNA internal transcribed spacer (ITS) sequences (Fig. 8b). USDA 122 and other *B. diazoefficiens* strains possessing 122-type *nopP* were found in the same cluster within the *B. diazoefficiens* clade, whereas *B. japonicum* strains possessing 122-type *nopP* were randomly distributed within the *B. japonicum* clade. Furthermore, the distribution of NopP types (110, 122, ST1, ST2) was not consistent with the bradyrhizobial lineage based on ITS sequences.

**Phosphorylation of NopP.** NopP from *S. fredii* NGR234 can be phosphorylated in vitro by plant kinases[31]. In addition, *B. elkanii* may secrete phosphorylated NopP, because several NopP spots with different pI were detected by two-dimensional gel electrophoresis of extracellular proteins[32]. To test whether USDA 122 secretes phosphorylated NopP, we analyzed extracellular proteins of USDA 122 in the culture by Phos-Tag SDS-polyacrylamide gel electrophoresis (PAGE) with or without the alkaline phosphatase treatment. The results showed that the alkaline phosphatase had no effect on the mobility of NopP, indicating that it was not phosphorylated (Supplementary Fig. 12).

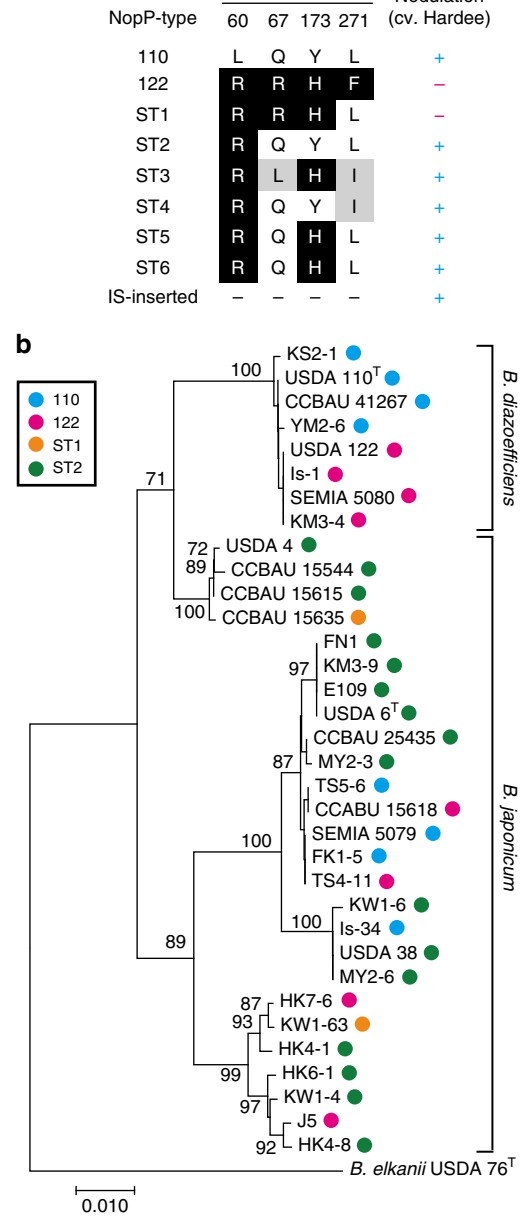

**Fig. 8** Diversity of NopP among soybean bradyrhizobia. **a** Differences in deduced NopP amino acid sequences among *B. diazoefficiens* and *B. japonicum* strains. Amino acid residues in NopP at positions 60, 67, 173, and 271 in each type and nodulation phenotype of a representative strain(s) on *G. max* cv. Hardee (Supplementary Fig. 11) are shown. **b** Phylogenetic tree of *B. diazoefficiens* and *B. japonicum* based on 16S–23 S rRNA internal transcribed spacer (ITS) sequences. Numbers at the nodes are the percentages of 1000 bootstrap replications supporting that partition (values ≥ 70% are shown). NopP types (indicated by colored circles) were assigned on the basis of the patterns of amino acid sequences deduced from genomic sequences (Supplementary Table 3). The scale bar shows the number of base substitutions per nucleotide

**Spontaneous mutations of *nopP* in Is-1 mutants.** The gene encoding 122-type NopP was found in *B. diazoefficiens* Is-1 (Supplementary Table 2), which is incompatible with *Rj2*-soybean plants[33]. Although, Tsurumaru et al.[34] previously conducted a screening by Tn5-based transposon mutagenesis and obtained eight *Rj2*-compatible mutants of Is-1, Tn5 insertions were located in genomic regions other than *rhc*, *ttsI*, and *nopP* genes[34]. Thus,

| Table 1 Symbiotic phenotypes of *V. radiata* cv. KPS1 inoculated with wild-type *B. diazoefficiens* strains and their *nopP* mutants | | | | |
|---|---|---|---|---|
| Inoculant | NopP sequence type | Nodule number $^a$ | Nodule dry mass (mg) $^a$ | Plant dry mass (mg) $^a$ |
| USDA 110$^T$ | 110 | 79.7 ± 4.3 | 63.6 ± 3.5 | 600 ± 26 |
| 110*nopP*$_{122}$ | 122 | 13.8 ± 2.4** | 35.9 ± 3.4** | 367 ± 18* |
| 110Δ*nopP* | Deleted | 61.3 ± 4.5** | 49.6 ± 5.5 | 481 ± 36 |
| USDA 122 | 122 | 2.4 ± 0.6 | 8.5 ± 2.4 | 292 ± 9 |
| 122*nopP*$_{110}$ | 110 | 27.0 ± 2.6** | 34.0 ± 3.4** | 359 ± 25** |
| 122Δ*nopP* | Deleted | 21.7 ± 1.5** | 32.9 ± 3.6** | 348 ± 26* |
| Uninoculated | – | 0 | 0 | 282 ± 9 |

$^a$Combined data from two independent experiments are shown. Values are mean ± SE ($n = 11$ for USDA 110$^T$, $n = 18$ for USDA 122, and $n = 12$ for the others). Asterisks (*) indicate significant differences from the wild-type strain (USDA 110$^T$ or USDA 122) by Dunnett's multiple comparison test (*$P < 0.05$, **$P < 0.01$)

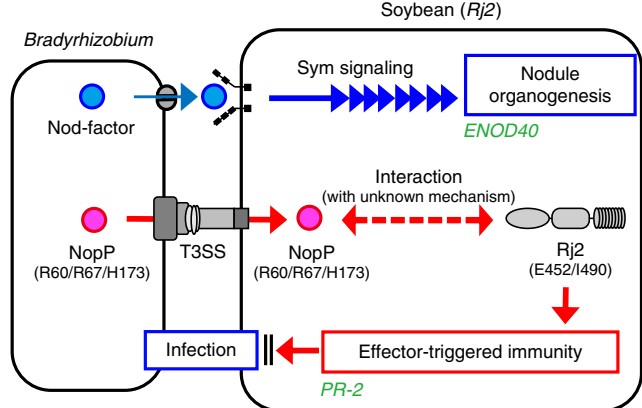

**Fig. 9** Proposed molecular mechanism of *Rj2* symbiotic incompatibility. Bradyrhizobia produce Nod factors to activate a signaling cascade (symbiosis [Sym] signaling) leading to nodule organogenesis. In the interaction with *Rj2*-soybeans, incompatible bradyrhizobia inject the NopP effector having the R60, R67, and H173 residues via T3SS into the host root cell. NopP interacts with the host Rj2 protein with unknown mechanism and activate effector-triggered immunity, which blocks the infection

we searched for spontaneous mutations of *nopP* in the Tn5 mutants[34]. Sanger sequencing revealed mutations caused by an IS element insertion (in the 7C2 mutant) or a nucleotide substitution (in 10C1) within the *nopP* coding region (Supplementary Table 4). These results suggest that suppression of *Rj2*-mediated incompatibility in at least these two mutants was due to spontaneous mutations of *nopP*, and that such mutations occur not only in USDA 122, but also in another *Rj2*-incompatible *Bradyrhizobium* strain.

**Effects of NopP on the symbiosis with mung bean**. To investigate whether NopP has a role in the interactions between rhizobia and other legumes, we inoculated wild-type strains of *B. diazoefficiens* and their *nopP* mutants on *Vigna radiata* (mung bean) cv. KPS1. Inoculation with the strains having the gene encoding 122-type NopP resulted in significantly fewer nodules and lower plant biomass than inoculation with the strains having the gene encoding 110-type NopP (Table 1). *Rj2* orthologs are conserved in the genome of various legumes including *V. radiata* (Supplementary Table 5). Therefore, symbiotic incompatibility mediated by NopP could also occur in leguminous plants other than soybean. In contrast, the plant inoculated with 110Δ*nopP* formed significantly fewer nodules and had a lower biomass than plants inoculated with wild-type USDA 110$^T$ (Table 1), suggesting that 110-type NopP, which induced no *Rj2*-mediated incompatibility, is advantageous for nodulation on *V. radiata* cv. KPS1.

## Discussion

Genotype-specific symbiotic incompatibility in legume–rhizobia interactions is an important trait for the use of root nodule bacteria to improve the crop yield[35]. Genetic loci that control root nodule symbiosis of soybean plants have been analyzed for more than 60 years[20]. Thanks to the recent remarkable progress in genomic and genetic technologies, the host plant *R*-genes and rhizobial T3SS (*rhc*) genes have been found to be responsible for the induction of symbiotic incompatibility[23,24,32]. These findings imply that host plants sense a specific rhizobial T3SS effector via an R protein, and reject rhizobial infection via a defense response[16]. However, the T3SS effector interacting with the host R protein, their interaction mechanism, and the signaling pathway that induces genotype-specific incompatibility have remained unknown.

Here we report the rhizobial determinant and propose the molecular mechanism of *Rj2* symbiotic incompatibility (Fig. 9).

Using spontaneous mutants of *B. diazoefficiens* USDA 122, we identified a rhizobial T3SS effector, NopP, as a causal factor for symbiotic incompatibility between bradyrhizobia and *Rj2*-genotype soybean plants (Figs. 2, 3). Our mutational and culture collection analyses of *nopP* revealed that three amino acid residues (R60, R67, and H173) in NopP are required for *Rj2*-mediated incompatibility (Figs. 4, 7, 8). Complementation analysis confirmed that *Rj2*-mediated incompatibility is associated with both bradyrhizobial NopP and host Rj2 protein (Fig. 5). Expression of a host defense marker gene *PR-2* was significantly increased following inoculation with the incompatible strains, whereas a symbiosis marker gene, *ENOD40*, was induced regardless of the compatibility of inoculants (Fig. 6). These results indicate that *Rj2*-soybeans activate symbiotic signaling process toward nodule organogenesis in response to incompatible rhizobia, but constantly monitor the specific variants of NopP and reject rhizobial infection by activating Rj2-mediated ETI (Fig. 9).

The first *nopP* homolog was identified as a host-inducible gene in *S. fredii* USDA 201[36]. Subsequent studies revealed that *nopP* encodes an effector protein secreted by T3SS and inactivation of *nopP* in *S. fredii* altered nodulation positively or negatively depending on the host plant[26,37]. Schechter et al.[38] demonstrated that *S. fredii* injects NopP into *Vigna unguiculata* nodule cells in a T3SS-dependent manner. In response to a flavonoid, *B. diazoefficiens* USDA 122 secretes NopP into culture supernatants through T3SS (Fig. 3), similar to other bradyrhizobia[27,32]. The above data suggest that bradyrhizobial NopP is injected via T3SS into plant cells.

The Rj2 protein belongs to the TIR-NBS-LRR class of R proteins. In plant–pathogen interactions, NBS-LRR proteins typically interact with corresponding avirulence (Avr) effector proteins to induce ETI[3], which is associated with the hypersensitive response (localized programmed cell death to restrict pathogen growth in plant cells). Most plant resistance proteins belong to the NBS-LRR superfamily directly or indirectly recognize pathogen's effectors to activate defense signaling[3,39]. Our results draw parallels between plant–pathogen and plant-mutualist interactions, because NopP acts as an Avr protein that is necessary for USDA 122 to trigger *Rj2*-dependent incompatibility. That incompatibility and compatibility differ by few amino acid residues in NopP protein, which is different from previous observations that epitopes in effector proteins have a more important role[39–41].

Rhizobial NopP is not predicted to contain any known functional domains and has no homology with any Avr effectors in pathogens, suggesting that NopP is evolved in symbiotic rhizobial systems. Our future study includes the analysis of direct interactions between those proteins and the screening for NopP-interacting host proteins to understand the recognition of NopP effector by the host R protein.

In general, ETI involves rapid synthesis of salicylic acid (SA) is rapidly synthesized as an essential signal molecule in response to Avr effector, subsequently inducing SA-dependent defense-signaling marker gene (e.g., *PR-1*, *PR-2*, and *PR-5*)[42,43]. In incompatibility response of *Rj4*-soybean plants with *B. elkanii* USDA 61, SA is also accumulated and *PR-1* expression was increased in roots[44]. Following inoculation with USDA 122, *PR-2* gene in *Rj2*-soybeans was transiently induced at an early stage of the infection (2 DAI, Fig. 6). The timing of *PR-2* induction was correlated with the inhibition of IT development in root hairs of Hardee (Fig. 4e). Therefore, the results suggest that *Rj2*-soybeans activate host defense response by inoculation with USDA 122 at an early stage of nodulation and downstream defense-signaling pathway may be different from *Rj4*-symbiotic incompatibility.

Homologs of *nopP* are present in rhizobia[26], but not in pathogens. All known *Bradyrhizobium* soybean symbionts possess *nopP* in the middle of the *nif* gene cluster (Fig. 3a, Supplementary Fig. 7a). These facts imply that NopP plays an important role in symbiotic nitrogen fixation. USDA 110$^T$ elicited significantly more nodules on a mung bean cultivar than did 110$\Delta nopP$ (Table 1), but this effect of 110-type *nopP* was not observed on either *Rj2*-soybean cv. Hardee (Fig. 7). Thus, USDA 110-type *nopP* in *Bradyrhizobium* strains likely has a positive role in nodulation in a host plant–dependent manner, which is consistent with the reports for *nopP* in *S. fredii*[31,37]. It is still unclear why USDA 122 secretes NopP that is disadvantageous for symbiosis with *Rj2*-soybeans. A possible explanation is that 122-type NopP may be advantageous for symbiosis with certain other plant genotypes.

Little is known about the importance and evolutionary aspects of genotype-specific symbiotic incompatibility between legumes and rhizobia. In plant–pathogen interactions, compatibility has been proposed to be the consequence of an evolutionary arms race[45,46]. As symbiotic compatibility changes as a result of substitutions of only a few residues in the host R protein[23] and rhizobial NopP (Fig. 9), *Rj2*-mediated incompatibility may be part of the evolutionary arms race to facilitate co-evolution between both partners. The mutants of USDA 122 (W3-1a, W9-1a) and Is-1 (7C2, 10C1) carried a mutated *nopP* and avoided the selection pressure from *Rj2*-soybeans (Fig. 2, Supplementary Table 4). Variation in this gene seems to have occurred independently in different bradyrhizobial lineages (Fig. 8). Thus, we suggest that bradyrhizobia often evolve *nopP* to circumvent the plant immune system. Furthermore, NopP-mediated incompatibility observed in a mung bean cultivar (Table 1) and *Rj2* orthologs are conserved in various legume species (Supplementary Table 5). These evidences suggest that determination of compatibility via NopP and R proteins is widely distributed in legumes with nodule symbiosis.

## Methods

**Bacterial strains and growth conditions**. Bacterial strains are listed in Supplementary Data 1. *Bradyrhizobium* strains were grown aerobically at 30 °C in HEPES-MES (HM) salt medium[47] (supplemented with 0.1% arabinose and 0.025% [wt/vol] yeast extract). *Escherichia coli* strains were grown at 37 °C in Luria–Bertani medium[48]. *Agrobacterium rhizogenes* was grown at 28 °C in Yeast Extract Peptone medium[49]. The following antibiotics were added: for *Bradyrhizobium* strains, kanamycin (Km) at 100 mg l$^{-1}$, spectinomycin (Sp) at 100 mg l$^{-1}$, and streptomycin (Sm) at 100 mg l$^{-1}$, or polymyxin B (Px) at 50 mg l$^{-1}$; for *E. coli*, ampicillin (Ap) at 50 mg l$^{-1}$, Sp at 25 mg l$^{-1}$, and Sm at 25 mg l$^{-1}$, or Km at 50 mg l$^{-1}$; for *A. rhizogenes*, chloramphenicol at 30 mg l$^{-1}$ or Km at 100 mg l$^{-1}$.

**DNA manipulations**. The isolation of plasmid DNA, restriction enzyme digestions, DNA ligations, and the transformation of *E. coli* were done as described by Sambrook and Russell[50]. Plasmids used in this study are listed in Supplementary Data 1. Oligonucleotide primers for PCR are listed in Supplementary Data 2.

**Plant growth conditions and nodulation assays**. The soybean cultivars used in the study were *G. max* (L.) Merr. cv. Hardee (*Rj2*), CNS (*Rj2*), IAC-2 (*Rj2*), and Lee (*rj2*). The mung bean cultivar was *V. radiata* cv. KPS1. For nodule counting and acetylene-reducing activity (ARA) assays, seeds of *G. max* and *V. radiata* were surface sterilized in 0.5% sodium hypochlorite for 1 min and then washed 10 times with sterile distilled water. The seeds were sown in a 300 ml plant box (CUL-JAR300; Iwaki, Tokyo, Japan) containing sterile vermiculite and watered with a nitrogen-free plant nutrient solution[51]. *Bradyrhizobium* strains were cultured for 5 or 6 days. The number of cells was adjusted to 10$^7$ ml$^{-1}$ in sterile water by direct counting with a Thoma hemocytometer (Kayagaki Irika Kogyo Co. Ltd., Tokyo, Japan), and 1 ml aliquots were inoculated onto surface-sterilized seeds. Plants were grown in a plant growth cabinet (NK Systems Co. Ltd., Osaka, Japan) at 25 °C with a photoperiod of 16 h light/8 h dark. The number of nodules and the dry weights of nodule and plant were determined 28 DAI. The ARA of roots was analyzed as described by Itakura et al.[52] Briefly, the roots were transferred into a 100 ml vials (SVG-100, Nichidenrika-glass, Co. Ltd., Kobe, Japan) and acetylene gas was injected at a final concentration of 10% (vol/vol). The vial was incubated for 30 min at 25 °C. The ethylene concentration was determined with a Shimadzu GC-18A gas chromatograph (Shimadzu Co., Kyoto, Japan) equipped with a flame ionization detector and a Porapack N column (GL Sciences, Inc., Tokyo, Japan).

For observation of early infection event and evaluation of gene expression in soybean roots, surface-sterilized Hardee seeds were germinated at 28 °C for 3 days on two-layered moistened kimwipes. Two seedlings were aseptically transferred to a seed pack (Daiki Rika Co. Ltd., Saitama, Japan) with nitrogen-free medium. One day after transplantation, the seedlings were inoculated with 1 × 10$^7$ cells (in 0.1 ml) of *B. diazoefficiens* strains or 0.1 ml sterilized water (uninoculated control). Plants were cultivated in a growth chamber at 25 °C and 70% humidity with a photoperiod of 16 h light/8 h dark. For observation of infection threads formed in root hair inoculated with a *gusA*-tagged strain, 2-day root samples were immersed in a β-glucuronidase staining solution [100 mM phosphate buffer (pH 7.0), 0.5 μM K$_3$(Fe(CN)$_6$), 0.5 μM K$_4$(Fe(CN)$_6$), 0.3% Triton X-100, and 200 mg ml$^{-1}$ of 5-bromo-4-chloro-3-indolyl-β-d-glucuronic acid]. The roots were subjected to vacuum treatment for 15 min and incubated overnight at 28 °C. The number of infection threads was counted with an Olympus BX51N Microscope (Olympus, Tokyo, Japan).

**Isolation of spontaneous mutants of USDA 122**. The wild-type *B. diazoefficiens* USDA 122 strain was inoculated onto surface-sterilized *G. max* cv. Hardee seeds and plants were cultivated as described above. After 21 days, nodules were picked from each root, dipped in 70% ethanol for 5 s, and surface sterilized in 1% (w/v) sodium hypochlorite for 5 min. Nodules were washed several times with sterile water and transferred into a 1.5 ml tubes (one nodule per tube) containing 100 μl of HM liquid medium. Each nodule was homogenized with a sterile toothpick and the suspension was streaked onto HM agar plates. After 7 days of incubation, single colonies that appeared from each nodule were transferred onto a new HM plate for purification.

**Whole-genome resequencing**. Total DNA was extracted from a stationary-phase culture of *Bradyrhizobium* cells by using an illustra bacteria genomicPrep Mini Spin Kit (GE Healthcare UK Ltd., Buckinghamshire, UK) and was processed using a Nextera DNA Sample Preparation Kit (Illumina, San Diego, CA, USA) to generate a shotgun library with unique index adapters. The library from each *Bradyrhizobium* strain was sequenced on a MiSeq system (Illumina), yielding ~250 bp paired-end reads. The raw reads from each *Bradyrhizobium* genome were trimmed and assembled de novo in CLC Genomic Workbench software (CLC bio, Inc., Aarhus, Denmark). The mutations in spontaneous mutants were identified by comparing the obtained MiSeq short reads and the reference *B. diazoefficiens* USDA 122 genome sequence (DDBJ/EMBL/GenBank accession number CP013127) using the "missing *k*-mer" tool of ShortReadManager[53,54].

**Western blotting and Phos-tag SDS-PAGE**. Rabbit polyclonal antibody to NopP protein (anti-NopP) was raised and affinity-purified against the synthetic peptide CIKGETFREKFGRND (GenScript Japan, Inc.,Tokyo, Japan). Extracellular proteins from the culture supernatants of *Bradyrhizobium* strains were isolated by phenol extraction[27] and separated (3 μg protein per well) by 12.5% SDS-PAGE. The separated proteins were transferred to PVDF membrane using a semi-dry transfer system (HorizeBLOT 4M-R, ATTO, Tokyo, Japan). The membranes were incubated with anti-NopP (diluted 1:10,000) for 30 min at room temperature. The membranes were then incubated with horseradish peroxidase-conjugated anti-rabbit IgG (Promega, Inc., Madison, WI, USA) diluted 1:2500 for 30 min, and the immunoreactive bands were detected using the chemiluminescent substrate,

Chemi-Lumi One Super (Nacalai Tesque, Inc., Kyoto, Japan). We confirmed that NopP (predicted molecular mass: 31 kDa) of USDA 122 was detected around 35 kDa (Supplementary Fig. 13). Phosphorylation of NopP protein was investigated using Phos-Tag SDS-PAGE technology, which was developed for mobility shift detection of phosphorylated proteins[55]. The polyacrylamide gels consisted of a Bis-Tris acrylamide gel copolymerized with Phos-tag [8% acrylamide, 20 μM acrylamide-pendant Phos-tag (Wako, Inc. Osaka, Japan), 40 μM ZnCl₂]. After electrophoresis, Phos-Tag gels were subjected to western blotting analysis as described above. Dephosphorization of extracellular proteins was performed according to modified procedures described by Kinoshita et al.[55]. Alkaline phosphatase (CIAP, Takara Bio, Inc., Kusatsu, Japan) was added to extracellular proteins (1 U μg$^{-1}$ protein) in appropriate buffer and incubated at 37 °C for 0, 30, 60, and 120 min. The dephosphorylation reactions were stopped by addition of a half-volume of 3 × SDS-PAGE loading buffer.

**Construction of *B. diazoefficiens nopP* mutants**. A mobilizable plasmid for generating in-frame *nopP* deletion mutants (122Δ*nopP* or 110Δ*nopP*) was constructed as follows: a 0.6 kb fragment containing the 5′-portion of the *nopP* coding region (+1 to +156 bp) and the upstream flanking region and a 0.7 kb fragment encoding the C-terminal region of *nopP* (+823 to +834 bp) and containing the 3′-flanking region were amplified by PCR. The two PCR products were fused by overlap extension PCR and the fused fragment was cloned into the SmaI site of pK18*mobsacB*[56]. The resulting plasmid (pMS130) was transferred from *E. coli* DH5α into *B. diazoefficiens* USDA 122 and 110$^T$ by triparental mating with the mobilizing strain *E. coli* HB101 harboring the pRK2013 helper plasmid[57]. A Km/Px-resistant and sucrose-sensitive transconjugant was selected for single-crossover insertion of the plasmid into the chromosome. The cells were grown in HM liquid medium and were spread on HM agar medium containing Px and 10% (w/v) sucrose to rescreen for sucrose-resistant colonies. The selected clones were further screened for Km sensitivity. Double-crossover events and an in-frame deletion of *nopP* were confirmed by PCR and Sanger sequencing in a 3730xl DNA Analyzer with a BigDye Terminator Cycle Sequencing Reaction Kit (Thermo Fisher Scientific, Inc., Waltham, MA, USA).

To swap *nopP* between USDA 122 and 110$^T$, the fragments containing 1.9 kb *nopP* and its flanking region were amplified by PCR from genomic DNA and cloned into the SmaI site of pK18*mobsacB*. The resulting plasmid pMS127 (containing *nopP* of USDA 122 [*nopP*$_{122}$]) or pMS128 (containing *nopP* of USDA 110$^T$ [*nopP*$_{110}$]) was transformed into USDA 110$^T$ or 122, respectively. The candidate double-crossover mutants (122*nopP*$_{110}$ or 110*nopP*$_{122}$) were selected as described above. *nopP* sequences were confirmed by Sanger sequencing. 122*nopP*_m3 was obtained as a by-product of constructing strain 122*nopP*$_{110}$.

**Construction of mutated *nopP* variants**. Sixteen 1.5 kb fragments containing the *nopP* coding and flanking regions and carrying the target nucleotide substitutions were obtained by overlap extension PCR with unique oligonucleotide primers (Supplementary Data 2). Each of these fragments was inserted into the SmaI site of pK18*mob*[56], yielding pMS155 to pMS170 (Supplementary Data 1). These plasmids were transferred into the *nopP* deletion mutant of USDA 110$^T$ (110Δ*nopP*) by triparental mating as described above. A Km/Px-resistant transconjugant was selected for single-crossover insertion of each plasmid into the chromosome, yielding 110Δ*nopP*$_{v1}$ to 110Δ*nopP*$_{v16}$. The single-crossover events were confirmed by PCR, and *nopP* sequences were verified by Sanger sequencing.

**Rj2 complementation test using hairy root transformation**. To obtain a cDNA fragment corresponding to the *Rj2* gene, total RNA was extracted from *G. max* cv. Hardee nodules formed by inoculation with *B. diazoefficiens* USDA 110$^T$ with a NucleoSpin RNA Plant Kit (Macherey-Nagel, Inc., Düren, Germany) according to the manufacturer's instructions. First-strand cDNA was synthesized using a SuperScript III First Strand cDNA Synthesis System (Thermo Fisher Scientific, Inc.) and the specific oligonucleotide primer Rj2-3UTR_R3. The first-strand cDNA was subjected to PCR with Rj2_Y2H_F2 and Rj2_Y2H_R3 primers to amplify the full-length *Rj2* cDNA fragment, and the product was subsequently cloned into the pENTR-dTOPO vector (Thermo Fisher Scientific, Inc.), yielding pMS145 as an entry clone for the Gateway Cloning System. To generate *rj2* cDNA, G1354 and T1469 (corresponding to E452 and I490 in the protein) in *Rj2* cDNA were substituted with A1354 and G1469 by site-directed mutagenesis using overlap extension PCR[58]. The DNA fragment obtained was cloned into the same vector, yielding pMS154 as an entry clone. cDNAs were transferred from the entry clones into the binary vector pUB-GW-GFP[28] between the polyubiquitin (*LjUbq1*) promoter and the *nos* terminator using Gateway LR Clonase II (Thermo Fisher Scientific, Inc.). These constructs were used to transfect *A. rhizogenes* K599. Six-day-old seedlings of *rj2*-soybean cv. Lee were used for complementation testing; *A. rhizogenes*-mediated hairy root transformation was based on the protocol described by Kereszt et al.[59] In brief, the cotyledonary node was infected with overnight cultures of K599 carrying *Rj2* cDNAs (E452/I490 or K452/R490) using a syringe. Infected seedlings were maintained in sterile vermiculite pots in a growth chamber under high humidity until hairy roots developed at the infection site (~2 weeks). The seedlings, with the main roots and non-GFP roots removed

under fluorescence binocular microscope, were cultured in sterile vermiculite pots containing 1/2 B&D solution[60] with 0.5 mM NH₄NO₃ for 1 week. The seedlings were inoculated with *B. diazoefficiens* USDA 122 (wild type or 122*nopP*$_{110}$) under nitrogen-free conditions[51] and nodulation on transgenic roots was examined 4 weeks after inoculation.

**Quantitative RT-PCR of genes in the roots of soybean**. Two plants were sampled from a seed pack at 2 or 4 DAI and the main roots were merged in a 15 ml tube. The roots were immediately frozen in liquid nitrogen and kept at −80 °C until RNA extraction. Total RNA was extracted using a NucleoSpin RNA Plant Kit (Macherey-Nagel, Inc.) and additionally treated with DNase I (Promega, Inc.) according to the manufacturer's instructions. First-strand cDNA was synthesized using a SuperScript III First Strand cDNA Synthesis System for reverse-transcription PCR (RT-PCR) (Thermo Fisher Scientific, Inc.) with random hexamers. Relative expression was analyzed by quantitative RT-PCR in a LightCycler Nano Instrument (Roche, Basel, Switzerland) using FastStart Essential DNA Green Master (Roche) and specific primers (Supplementary Data 2). The PCR reaction mixture was adjusted according to the manufacturer's instructions and thermal cycling conditions consisted of 10 min at 95 °C, and 45 cycles of 20 s at 95 °C, 20 s at 55 °C, and 20 s at 72 °C. The specificity of PCR amplification was confirmed by a melting-curve analysis. Transcript levels were normalized to the expression of *SUBI2* (ubiquitin) measured in the same samples[61].

***nopP* diversity and phylogenetic analysis**. Total DNA was extracted from the cells of *Bradyrhizobium* strains in the Japanese culture collection[30] by using an illustra bacteria genomicPrep Mini Spin Kit (GE Healthcare UK, Ltd.). *nopP* with its flanking regions was amplified by PCR with the oligonucleotide primers nopP_F1 and nopP_R1. Purified PCR products were subjected to Sanger sequencing with the primers nopP_F1, nopP_R1, nopP_F2, and nopP_R3. The obtained sequences were assembled and amino acid sequences were deduced in Genetyx-MAC v. 18.0.3 software (Genetyx Co., Tokyo, Japan). The ITS sequences between the 16S and 23S rRNA genes for phylogenetic analysis are listed in Supplementary Table 2. The sequences were aligned using the CLUSTALW program[62]. Neighbor-joining trees were constructed in MEGA v. 7 software[63] and 1000 bootstrap replicates were used to generate a consensus tree.

**Statistical analysis**. Statistical significance was determined using a two-tailed Student's *t*-test performed by Microsoft Excel for pairwise comparisons. A one-way analysis of variance followed by a Dunnett's test or by Tukey's honest significant difference test was performed using R 3.3.2 software (https://www.r-project.org/) with "multcomp" package for comparisons of multiple test samples. For non-normal distribution data, significance of difference among groups was evaluated by Steel–Dwass test using JMP software (SAS Institute Inc. Cary, NC, USA). *P*-values < 0.05 were considered statistically significant. Sample size ($n$) used for experiments is indicated in the figure legends.

**Data availability**. MiSeq sequences were deposited in the DDBJ Sequence Read Archive DRA005835 [DRX086799 to DRX086806] and DRA005773 [DRX085569]. DNA sequences of the *nopP* genes determined by the Sanger method were deposited in the GenBank/EMBL/DDBJ under the accession codes (LC331619 to LC331646, and LC331679 to LC331683) listed in Supplementary Table 2. All relevant data that support the findings of this study are available from the corresponding author upon reasonable request.

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

## Acknowledgements

We thank Professor Neung Teaumroong (Suranaree University of Technology) for providing mung bean seeds. We thank Dr. Cristina Sánchez and Mr. Shohei Kusakabe (Tohoku University) for advice on mutant construction and protein secretion assay, respectively. We are grateful to Ms. Kaori Kakizaki (Tohoku University) for technical supports in MiSeq sequencing of the spontaneous mutants, and to Mr. Tomoyuki Sakata (Kyushu University) for technical support in DNA sequencing of Is-1 mutants. We thank Dr. Michiko Yasuda (Tokyo University of Agriculture and Technology) and Dr. Yasuyuki Kawaharada (Iwate University) for advice on microscopic observation of soybean roots. This work was supported by JSPS KAKENHI Grant Numbers 26252065 (to K.M.) and 15K20868 (to M.S.) from the Ministry of Education, Culture, Sports, Science and Technology of Japan.

## Author contributions

M.S., H.I., and H.O. carried out screening, resequencing, and genetic validation of USDA 122 spontaneous mutants. M.S. constructed mutant strains and *nopP* variants of *B. diazoefficiens*. M.S., S.T., Y.S., and H.K. participated in inoculation tests of *B. diazoefficiens* and its mutants. M.S. and Y.K. carried out gene expression analysis of soybean plants. Y.U. carried out the *Rj2/rj2* complementation test. M.S. and S.T. sequenced *nopP* and constructed the phylogenetic tree. H.T. and T.Y. sequenced *nopP* in Tn5 mutants of *B. diazoefficiens* Is-1. M.S., S.S., H.M., and K.M. conceived and coordinated experiments. M.S., Y.U., and K.M. wrote the manuscript.

## Additional information

**Competing interests:** The authors declare no competing interests.

