## [Peer Review File · Nature Communications]

Reviewer #1 (Remarks to the Author):

The soybean Rj2 gene encodes an NBS-LRR protein that confers resistance to nodulation by specific rhizobial strains. The Rj2 function is dependent on the bacterial type III secretion system; as such, Rj2 presumably recognizes a bacterial effector to trigger immune responses but such effectors have not been identified. This manuscript describes the identification and validation of NopP as the cognized effector of Rj2. The manuscript is generally well written but still needs significant improvements. The data presented are solid and strongly support the conclusion. However, in my opinion, the novelty of the research does not warrant its publication in Nature Communications. It is more appropriate for specialized journals such as Molecular Plant-Microbe Interactions (MPMI).

Minor comments:

1. In the abstract, lines 19-20: "Genotype-specific incompatibility in legume–rhizobium symbiosis has been suggested to control by effector-triggered immunity underlying pathogenic host-bacteria interactions." Multiple mechanisms could contribute to determining symbiosis specificity and this is just one of them. "to control" should be "to be controlled."
2. Lines 26-27: "The incompatibility conferred by NopP variants occurred through complementation of the host Rj2 gene in rj2-soybean." Please consider rephrasing the sentence. Complementation of rj2-soybean by the Rj2 allele...
3. Lines 32-33: "Rhizobia symbiotically fix atmospheric nitrogen in root nodules of leguminous plants, which thus acquire fixed nitrogen under nitrogen-limited conditions." This is not a good sentence.
4. Line 38: (e.g., NFR1 and NFR5 in *Lotus japonicus*).
5. Lines 50-42: "Flavonoids derived from host plants induce NodD...". Based on my understanding, nodD genes are constitutively expressed and NodD proteins are activated by flavonoids.
6. Lines 68-69: "suggesting that a T3SS-dependent effector protein in USDA 122 blocks nodulation with Rj2-soybean plants." "blocks" should be "block".
7. I don't like the term "Rj2 incompatibility" but prefer "Rj2-mediated incompatibility".
8. Lines 135-137: "The same symbiotic phenotype was observed in other Rj2-soybean cultivars, CNS and IAC-2, but not in a rj2-soybean cultivar Lee, inoculated with 122 Δ nopP." Why not the cultivar Lee? Mutation of nopP should not significantly change the compatible interaction (maybe I missed something?).
9. Lines 143-146: "Moreover, the NopP protein is secreted from USDA 110 cells via T3SS when genistein is added to culture medium, suggesting that symbiotic compatibility between bradyrhizobia and Rj2 soybeans is not determined by secretion of NopP." I believe the authors try to say that the compatibility was not determined by whether the effector is secreted but by the sequence of the secreted effector.
10. Lines 248-252: Does the mung bean cultivar carry a putative Rj2 ortholog?

11.Line 309: Did the authors try other methods to test the interaction between Rj2 and NopP? Is the statement of “data not shown” allowed in this journal?

Reviewer #2 (Remarks to the Author):

The manuscript entitled “Variation of three amino acid residues in bradyrhizobial NopP effector determines symbiotic incompatibility with Rj2-soybeans” by Minamisawa et al. is well written and very nice contribution to the field of plant-microbe interactions.

I believe that the paper could ultimately be a very nice publication in Nature Communications but is currently lacking some critical pieces of additional evidence, which I have listed below and then expand upon in some of the specific comments below.

1. The authors interpret their results as suggesting that certain residues in the NopP protein are critical for Rj2 recognition. However, their results would also be consistent with the fact that these amino acid changes could disrupt secretion of the NopP protein from the bacterium to the plant. Ideally, it would be good to actually see movement of the protein into the plant, which is now readily possible using methods pioneered in plant pathogens. However, short of this, the authors need to at least show that the various forms of NopP do not vary with regard to the ability of the bacterium to deliver the protein to the culture supernatant.

2. The authors interpret their results as indicative of a plant immune response being triggered via NopP-Rj2 recognition. However, they present no data in the paper that such a defense response does occur and, most importantly, is dependent on NopP-Rj2 recognition. These data are necessary if the authors on going to continue to claim a plant pathogen-type model for their results.

More specific comments:

p. 6, top: It would be nice to at least see more microscopy on these USDA122 spontaneous mutants. The varying fixation phenotypes suggests that the mutants may be blocked at differing steps in the symbiosis, which would be useful information to better understand how Rj2 restriction is mediated. Note that, although R genes are well studied, to date there has been no clear mechanism attributed to how R genes actually confer resistance. Perhaps this rhizobial system might provide better information.

p. 10, bottom: There appears to be an apparent inconsistency in the results that needs some mention in the paper. Specifically, work with USDA122 indicated that NopP residues R60 and H173 were the most critical, with R67 also playing a lesser role. The authors used these data to suggest

that these were critical residues for Rj2 recognition. However, in their studies of NopP in the ST3, ST5, and ST6 strains that obtain an opposite result, NopP strains having the R60 and H173 residues nodulated normally, while those with R67 did not nodulate. The authors cannot have it both ways. Either R60 and H173 are critical for Rj2 recognition or they are not. Some explanation is required.

p. 11, top: I find the results mentioned here interesting. I would ask that the authors compare the NopP lineage to that obtained by comparing the *nifHDK* and *nodABC* genes. The latter differ from one another and also do not follow the rRNA phylogeny. It would be interesting to see if *nopP* reflects co-evolution with either the nitrogen fixation or nodulation genes. (see also discussion on p. 15). The location of NopP within the *nif* cluster suggests co-evolution with the *nif* genes, which seems counter-intuitive if the primary role of NopP is to regulate the nodulation process and not nitrogen fixation. Hence, I would again underline the need for the authors to better define the infection blockage imposed by Rj2 in relation to NopP-determined specificity. One cannot rely solely on the published results of others.

p. 14, line 300: The suggestion that the R and H residues in NopP are the sites of phosphorylation is a bit too much speculation, especially considering that relatively easy experiments could be done to look at this.

p. 14, line 308: Y2H is notorious for both false positives and false negatives and, hence, mentioning these data seems irrelevant.

p. 15, bottom: What we now know about R genes-effector interactions and the associated complexity far exceeds the older 'zigzag' model. Hence, I suggest that authors cite a more recent review paper that more adequately describes the extant knowledge in this area.

Figure 8 should be modified to make clear that the interaction, perhaps better stated as recognition, of NodP by Rj2 could be direct (as currently drawn in the figure) or indirect.

I found it interesting that, in the case of 122Ωrhcl (no T3SS occurs at all), nodule formation and plant growth were almost indistinguishable from USDA 110T. In other words, a variety of other effectors delivered from USDA 122 to Rj2-soybean are not necessary for nodule formation in Rj2-soybean? This is quite a different case than one would find in plant pathogens, for example, and, hence, deserves some discussion in the paper.

Regarding Fig. 2.

The authors need to show a picture of the nodule in the root.

Regarding Fig. 3.

Bradyrhizobium diazoefficiens NopP (244 amino acid) is predicted as approximately 27 kDa. However, in the Fig 3C, NopP was detected at the 35 kDa. So, NopP is regulated by post-translational gene regulation in Rj2 soybean? This is a particularly problematic part of the paper, especially considering how the data are subsequently discussed (see comment above). A few very easy

experiments could be done to help clarify this. For example, does phosphatase treatment shift the band on the gel so that it runs closer to 27 kDa?

122 Δ nopP showed more nodule number per plant than 122 Ω rhcJ (no T3SS occurs at all). Because of other effectors? Explain why?

Regarding Fig. 4.

The authors should confirm that the expression and secretion (see comment) above of the various NodP forms in either USDA 122 or USDA 110T are equivalent.

In the figure legend, Scale bars, 1 cm.  Scale bar, 1 cm.

Regarding Fig. 5.

The authors should also show GFP photographs in Lee + rj2 and Lee vector control.

It would be useful to demonstrate downstream gene expression to determine whether a defense response does occur upon treatment with USDA 122 or 122nopP110 in Lee + Rj2 and Lee + rj2, respectively. This might be the most direct way to document that plant immunity is indeed playing a role.

Mistakes

Line 117 (Fig. 1c)  (Fig. 2c)

Line 119 (Fig. 1c)  (Fig. 2c)

Reference 1

Bohlool, B. B., Ladha, J. K., Garrity, D. P. & George, T. Biological Nitrogen-Fixation for Sustainable Agriculture - a Perspective. *Plant Soil* 141, 1-11 (1992).

Bohlool, B. B., Ladha, J. K., Garrity, D. P. & George, T. Biological nitrogen fixation for sustainable agriculture : A perspective. *Plant Soil* 141, 1-11 (1992).

Reference 2

Wagner, S. C. Biological Nitrogen Fixation. *Nature Education Knowledge* 3, 15 (2012).

Wagner, S. C. Biological Nitrogen Fixation. Nature Education Knowledge 3, 15 (2011).

Reference 3

Dénarié, J. & Cullimore, J. Lipo-Oligosaccharide Nodulation Factors - a Minireview New Class of Signaling Molecules Mediating Recognition and Morphogenesis. Cell 74, 951-954 (1993).

?

Dénarié, J. & Cullimore, J. Lipo-Oligosaccharide Nodulation Factors : A New Class of Signaling Molecules Mediating Recognition and Morphogenesis. Cell 74, 951-954 (1993).

Reference 19

van Berkum, P. & Fuhrmann, J. J. Evolutionary relationships among the soybean bradyrhizobia reconstructed from 16S rRNA gene and internally transcribed spacer region sequence divergence. Int J Syst Evol Microbiol 50 Pt 6, 2165-2172 (2000).

?

van Berkum, P. & Fuhrmann, J. J. Evolutionary relationships among the soybean bradyrhizobia reconstructed from 16S rRNA gene and internally transcribed spacer region sequence divergence. Int J Syst Evol Microbiol 50, 2165-2172 (2000).

Reference 25

Sugawara, M. et al. Complete Genome Sequence of Bradyrhizobium diazoefficiens USDA 122, a Nitrogen-Fixing Soybean Symbiont. Genome Announc 5, e01743-01716 (2017).

?

Sugawara, M. et al. Complete Genome Sequence of Bradyrhizobium diazoefficiens USDA 122, a Nitrogen-Fixing Soybean Symbiont. Genome Announc 5, e01743-16 (2017).

Reference 25

Cieśła, J., Fraczyk, T. & Rode, W. Phosphorylation of basic amino acid residues in proteins: important but easily missed. Acta Biochim Pol 58, 137-148 (2011).

?

Cieśła, J., Fraczyk, T. & Rode, W. Phosphorylation of basic amino acid residues in proteins: important but easily missed. Acta Biochim Pol 58, 137-147 (2011).

Reference 42

Kim, H. S. et al. The *Pseudomonas syringae* effector AvrRpt2 cleaves its C-terminally acylated target, RIN4, from Arabidopsis membranes to block RPM1 activation. *Proc Natl Acad Sci USA* 102, 6496-6501 (2005).

☐

Kim, H. S. et al. The *Pseudomonas syringae* effector AvrRpt2 cleaves its C-terminally acylated target, RIN4, from Arabidopsis membranes to block RPM1 activation. *Proc Natl Acad Sci USA* 102, 6496-6501 (2005).

In the many references have wrong lowercase or capital letter or italic.

Reviewer #3 (Remarks to the Author):

In this manuscript by Sugawara and colleagues, the authors identified NopP as an avirulence protein that is necessary for USDA122 to trigger Rj2-dependent incompatibility. Comparisons between NopP alleles suggested that compatibility and incompatibility is due to variation in one three amino acids.

This work is amazing! I am exceptionally enthusiastic about this manuscript because the work is very impactful. It draws parallels between mechanisms of plant-pathogen and plant-mutualist interactions and will definitely influence the thinking in the field. The data suggest that plant immunity may not distinguish beneficial from detrimental symbiont; NopP likely dampens PTI. The data further suggest that plant immunity has key nodes that are susceptible to attack by microbe-associated effector proteins. I also think it is possible that if Rj2 is a guard, that NopP and a pathogen effector protein likely modify in the same manner, the same guardee. That incompatibility and compatibility differ by few amino acids in NopP is novel; in plant-pathogen interactions, it is typically a presence/absence polymorphism in type III effector genes. Last, the results are wholly consistent with the conclusion that type III effector genes of rhizobia exhibit mutualistic co-evolution with host defenses.

The work is simple, yet brilliant. The experiments were logical and comprehensive and the results are convincing. The manuscript is tight and easy to follow. The authors smartly relied on spontaneous mutants and the easily visualized gain of nodulation phenotype to identify causative mutations in NopP. They engineered strains and also relied on natural variation in nopP alleles to

test their hypothesis. Sugawara and colleagues confirmed the gene-for-gene relationship between nopP and Rj2. Last, the authors even cleared up a previous study and showed that NopP has beneficial effects in mung bean.

However, my review is not without some minor comments. I felt the writing needs to be improved in three areas. First, in key places, there needs to be more attention to precision in language. Second, there are just enough errors and awkwardly phrased sentences to be noticeable. Third, the impact could be elevated if the authors were to discuss the broader implications of their work. I am a little worried that the discussion focuses too much on the specific NopP-Rj2 system and does not adequately relate the work to the larger implications of immunity-effector interactions in plant-microbe interactions. The impact of the work may not be sufficiently recognized by members of the wider field. Below are some, but not all, cases:

Precision:

Line 128: secretion is not sufficient to call it an effector. It is safer to say that NopP is secreted in a T3SS-dependent manner.

Line 135: did not test nitrogen fixation (strike these words out).

Line 138: change “induces” to “is necessary for”

Line 237: this is not clear to me. I am guessing that the authors sequenced the nopP alleles of the Is-1 mutants to ask if there were second site mutations in nopP?

Errors/awkward:

Line 18

Line 32

Line 43: escape PTI, not the patterns.

I am very supportive of this manuscript. I loved it.

Dear Reviewers,

We are grateful for your critical comments and useful suggestions that have helped us to improve our paper. As indicated in the responses that follow, we have taken all these comments and suggestions into account in preparing the revised version of our paper. All the points revised according to the reviewer comments were highlighted as red letters in the revised manuscript.

Reviewer #1 (Remarks to the Author):

Comment: The soybean *Rj2* gene encodes an NBS-LRR protein that confers resistance to nodulation by specific rhizobial strains. The *Rj2* function is dependent on the bacterial type III secretion system; as such, *Rj2* presumably recognizes a bacterial effector to trigger immune responses but such effectors have not been identified. This manuscript describes the identification and validation of NopP as the cognized effector of *Rj2*. The manuscript is generally well written but still needs significant improvements. The data presented are solid and strongly support the conclusion. However, in my opinion, the novelty of the research does not warrant its publication in Nature Communications. It is more appropriate for specialized journals such as Molecular Plant-Microbe Interactions (MPMI).

Response: Although genetic loci of soybeans restricting nodulation with specific rhizobial strains were found as *Rj2*, *Rj3* and *Rj4* in 1960's, the molecular mechanism of this genotype-specific incompatibility have been not fully elucidated so far. In particular, *Rj2*-mediated incompatibility has been strongly expected to reveal the crucial role of plant immunity of rhizobium-legume symbioses (Hayashi *et al.* 2012, Gourion *et al.* 2014), because *Rj2* gene encodes a typical R protein to counter the virulence functions of pathogen effector (Yang *et al.* 2010). As compared to our understanding of plant pathogen interactions, much remains to be elucidated for roles of effector-triggered immunity (ETI) in symbiosis. The present work clearly shows that the variations of NopP effector protein secreted by the incompatible strains determine *Rj2*-mediated incompatibility via ETI. According to additional experiments recommended by reviewers, we obtained two key results: By incompatible strains, *Rj2*-soybean plant activated a defense gene, *PR-2*, and suppressed infection thread number, 2 days after inoculation. By the solid and data, we are able to demonstrate the framework that *Rj2*-soybeans monitor the specific variants of NopP and reject bradyrhizobial infection via ETI mediated by the host *Rj2* protein. We believe that our results could provide important contributions towards our understandings for plant immunity between symbiosis and pathogenesis. These are reasons that we submit our manuscript to Nature Communications.

Minor Comment 1: In the abstract, lines 19-20: "Genotype-specific incompatibility in legume–rhizobium symbiosis has been suggested to control by effector-triggered immunity underlying pathogenic host-bacteria interactions." Multiple mechanisms could contribute to determining symbiosis specificity and this is just one of them. "to control" should be "to be controlled."

Response: We agree with your suggestion. This has been corrected as indicated.

Minor Comment 2: Lines 26-27: "The incompatibility conferred by NopP variants occurred through complementation of the host *Rj2* gene in *rj2*-soybean." Please consider rephrasing the sentence. Complementation of *rj2*-soybean by the *Rj2* allele...

Response: According to the reviewer comment, this has been rephrased to "Complementation of *rj2*-soybean by the *Rj2* allele conferred the incompatibility induced by USDA 122-type NopP."

Minor Comment 3: Lines 32-33: "Rhizobia symbiotically fix atmospheric nitrogen in root nodules of leguminous plants, which thus acquire fixed nitrogen under nitrogen-limited conditions." This is not a good sentence.

Response: The sentence has been re-written in the revised manuscript.

Minor Comment 4: Line 38: (e.g., NFR1 and NFR5 in *Lotus japonicus*).

Response: This has been corrected as indicated.

Minor Comment 5: Lines 50-42: “Flavonoids derived from host plants induce NodD...”. Based on my understanding, nodD genes are constitutively expressed and NodD proteins are activated by flavonoids.

Response: We agree with your suggestion. This has been corrected based on the reviewer comment.

Minor Comment 6: Lines 68-69: “suggesting that a T3SS-dependent effector protein in USDA 122 blocks nodulation with Rj2-soybean plants.” “blocks” should be “block”.

Response: We do not agree with the suggestion, because “a T3SS-dependent effector protein” is a single noun.

Minor Comment 7: I don’t like the term “Rj2 incompatibility” but prefer “Rj2-mediated incompatibility”.

Response: All of this term has been corrected to “Rj2-mediated incompatibility”.

Minor Comment 8: Lines 135-137: “The same symbiotic phenotype was observed in other Rj2-soybean cultivars, CNS and IAC-2, but not in a rj2-soybean cultivar Lee, inoculated with 122 Δ nopP.” Why not the cultivar Lee? Mutation of nopP should not significantly change the compatible interaction (maybe I missed something?).

Response: We are sorry for the confusion. This was a shortage of our explanation. The sentence was re-written to “The efficient symbiotic phenotype with 122 Δ nopP was also observed in other Rj2-soybean cultivars, CNS and IAC-2 (Supp. Fig. 6a–d). In a rj2-soybean cultivar Lee, the symbiotic phenotypes were not different between inoculations of USDA 122 and 122 Δ nopP (Supp. Fig. 6e, f).”

Minor Comment 9: Lines 143-146: “Moreover, the NopP protein is secreted from USDA 110 cells via T3SS when genistein is added to culture medium, suggesting that symbiotic compatibility between bradyrhizobia and Rj2 soybeans is not determined by secretion of NopP.” I believe the authors try to say that the compatibility was not determined by whether the effector is secreted but by the sequence of the secreted effector.

Response: We agree with the suggestion. The sentence has been rephrased as indicated in the revised manuscript.

Minor Comment 10: Lines 248-252: Does the mung bean cultivar carry a putative Rj2 ortholog?

Response: We have not been cloned cDNA of Rj2 ortholog in cultivar KPS1. Instead of this, we found Rj2 ortholog with high homology in the genome sequence of a cultivar of mung bean. We have added the results of BLAST search in Supplementally Table 4 and a sentence in the text.

Minor Comment 11: Line 309: Did the authors try other methods to test the interaction between Rj2 and NopP? Is the statement of “data not shown” allowed in this journal?

Response: The statement of “data not shown” is not allowed in this journal. In addition, the results of Y2H is not sufficient to say that Rj2 and NopP was interacted indirectly, since it is well-known that false negatives are detected by Y2H. We are just trying to purify recombinant Rj2 and NopP proteins and will analyze direct interactions between the proteins by Isothermal titration calorimetry. Therefore, we deleted the Y2H results from the discussion section, and we have re-written that the physical interaction mechanism of Rj2/NopP is still unknown.

Reviewer #2 (Remarks to the Author):

The manuscript entitled “Variation of three amino acid residues in bradyrhizobial NopP effector determines symbiotic incompatibility with Rj2-soybeans” by Minamisawa et al. is well written and very nice contribution to the field of plant-microbe interactions.

I believe that the paper could ultimately be a very nice publication in Nature Communications but is currently lacking some critical pieces of additional evidence, which I have listed below and then expand upon in some of the specific comments below.

Comment 1: The authors interpret their results as suggesting that certain residues in the NopP protein are critical for Rj2 recognition. However, their results would also be consistent with the fact that these amino acid changes could disrupt secretion of the NopP protein from the bacterium to the plant. Ideally, it would be good to actually see movement of the protein into the plant, which is now readily possible using methods pioneered in plant pathogens. However, short of this, the authors need to at least show that the various forms of NopP do not vary with regard to the ability of the bacterium to deliver the protein to the culture supernatant.

Response: Thank you for this valuable comment. We agree with your suggestion. According to your suggestion, we examined NopP secretion by USDA 110 and 110*nopP*₁₂₂ (USDA110 strain having USDA122-type *nopP*) to the culture by western blotting. The results have shown that both strains secreted NopP into the culture supernatants. Thus, we suggested that substitutions of NopP on the amino acid residues at positions 60, 67, 173, and 271 does not affect the ability of the bacterium to deliver the protein. We have included this data in Figure 4a, and re-written the sentence in the revised manuscript. In addition, we are now trying to construct a USDA122 mutant, which is able to secrete CyaA-fused NopP. We would like to examine the movement of CyaA-NopP into the roots of Hardee by measurement of Adenylate cyclase activity.

Comment 2. The authors interpret their results as indicative of a plant immune response being triggered via NopP-Rj2 recognition. However, they present no data in the paper that such a defense response does occur and, most importantly, is dependent on NopP-Rj2 recognition. These data are necessary if the authors on going to continue to claim a plant pathogen-type model for their results.

Response: Based on the reviewer comment, we examined the expressions of defense marker genes (*PR-1*, *PR-2* and *PR-5*) in Hardee (Rj2) inoculated with USDA 122 and 122*nopP*₁₁₀. The expression of *PR-2* was significantly induced 2 days after inoculation with USDA 122 as compared with 122*nopP*₁₁₀ and uninoculated control. Together with the results of Rj2/rj2 complementation analyses (Figure 5), our data suggested that NopP/Rj2-dependent incompatibility occurs via host defense response. We have added the procedure, results and discussions to the revised manuscript.

More specific comments:

Comment 3: p. 6, top: It would be nice to at least see more microscopy on these USDA122 spontaneous mutants. The varying fixation phenotypes suggests that the mutants may be blocked at differing steps in the symbiosis, which would be useful information to better understand how Rj2 restriction is mediated. Note that, although R genes are well studied, to date there has been no clear mechanism attributed to how R genes actually confer resistance. Perhaps this rhizobial system might provide better information.

Response: We agree with the reviewer comment. We also think that this rhizobial system may provide mechanistic understanding of R gene. However, the lacking of nitrogen fixation activity was caused by spontaneous mutation (deletion) of *nif* genes (Supp. Fig. 2 to 5). Therefore, we believe that Fix- mutants are not blocked at different stages in terms of nodulation processes.

Comment 4: p. 10, bottom: There appears to be an apparent inconsistency in the results that needs some mention in the paper. Specifically, work with USDA122 indicated that NopP residues R60 and H173 were the most critical, with R67 also playing a lesser role. The authors used these data to suggest that these were critical residues for Rj2 recognition. However, in their studies of NopP in the ST3, ST5, and ST6 strains that obtain an opposite result, NopP strains having the R60 and H173 residues nodulated normally, while those with R67 did not nodulate. The authors cannot have it both ways. Either R60 and H173 are critical for Rj2 recognition or they are not. Some explanation is required.

Response: Thank you for your comments. To evaluate the effects of R67 on nodulation of *Rj2*-soybeans, we performed two additional experiments: (i) When 110nopP_v1, _v2, _v4 and _v16 were inoculated to Hardee (*Rj2*) with 10 replications for each inoculant, the nodule numbers (per plant) of 110nopP_v2 (RRH) were significantly lower than those of 110nopP_v4 (RQH) ($P=0.022$) (Fig. 7, Supp. Fig. 10a in the revised manuscript). (ii) When *122nopP_m3* (RQH), a byproduct during swapping mutant construction of *nopP* genes between USDA 122 and USDA 110, were inoculated to Hardee (*Rj2*), the inoculation with *122nopP_m3* (RQH) showed apparently intermediate nodule numbers as compared with *122nopP_110* (LQY) and USDA 122 (RRH), although there were variations in F271 and L271 of NopP residue in the tested strains (Supp. Fig. 10b). These results strongly suggested that NopP residues R60, R67 and H173 are all critical to express full nodulation inhibition equivalent to USDA 122 inoculation, although the mutants having R60 and R173 showed slightly different nodulation phenotypes according to genomic backgrounds of USDA 122 and 110 (Suppl. Fig. 10). This is, thus, consistent with the results of ST3, ST5, and ST6 strains (Suppl. Fig. 11). We edited text including new data (Fig. 7, Supp. Fig. 10).

Comment 5: p. 11, top: I find the results mentioned here interesting. I would ask that the authors compare the NopP lineage to that obtained by comparing the *nifHDK* and *nodABC* genes. The latter differ from one another and also do not follow the rRNA phylogeny. It would be interesting to see if *nopP* reflects co-evolution with either the nitrogen fixation or nodulation genes. (see also discussion on p. 15). The location of NopP within the *nif* cluster suggests co-evolution with the *nif* genes, which seems counter-intuitive if the primary role of NopP is to regulate the nodulation process and not nitrogen fixation.

Response: According to the comment, we compared the phylogenetic trees based on *nopP* and the concatenated sequences of *nif* or *nod* genes. As the reviewer expected, *nif* genes lineage is more consistent with that of *nopP*, than phylogeny of *nod* genes or ITS, suggesting that *nopP* co-evolved with *nif* genes. We have added the descriptions in procedure, results (Supp. Fig. 13) and discussion sections in the revised manuscript. The finding implies a positive relationship of NopP to symbiotic nitrogen fixation, but we do not have any experimental evidence so far. We would like to clarify the function of NopP with this viewpoint in the future.

Comment 6: Hence, I would again underline the need for the authors to better define the infection blockage imposed by *Rj2* in relation to NopP-determined specificity. One cannot rely solely on the published results of others.

Response: According to the comment, we microscopically observed early infection events, such as colonization on the roots, root hair curling and infection thread (IT) formation on the roots of Hardee (*Rj2*) inoculated with USDA 122, USDA110^T and their *nopP* swapping mutants. We have found that the numbers of ITs formed on the roots inoculated with incompatible strains were significantly less than those with compatible strains (*122nopP_110* or USDA 110^T) at 2 days after inoculation. According to these results, *Rj2*-soybeans mainly block very early infection events of the incompatible rhizobia (e.g. rhizobial colonization on root hair tips and IT formation). Based on the results, we have included the data in the revised manuscript as Figure 4e and Supp. Fig. 8 and rewrote the Methods, Results, and Discussion sections.

Comment 7: p. 14, line 300: The suggestion that the R and H residues in NopP are the sites of phosphorylation is a bit too much speculation, especially considering that relatively easy experiments could be done to look at this.

Response: According to the reviewer's comment, we have done an additional experiment to know whether USDA 122 secrete phosphorylated NopP by Phos-Tag SDS-PAGE followed by Western blotting using anti-NopP. However, we could not detect phosphorylated-NopP in the culture using the methods, suggesting that the R and H residues in NopP of USDA 122 are not phosphorylated, in at least rhizobial cells. Based on the results, we have added the procedure and results to the revised manuscript and deleted the related discussions.

Comment 8: p. 14, line 308: Y2H is notorious for both false positives and false negatives and, hence, mentioning these data seems irrelevant.

Response: We agree with the reviewer's suggestion. It is not sufficient to mention that Rj2 and NopP is indirectly interacted by only the results of Y2H. Therefore, we deleted the description about Y2H, and we decided to say that the mechanism of physical interaction between Rj2 and NopP is still unknown.

Comment 9: p. 15, bottom: What we now know about R genes-effector interactions and the associated complexity far exceeds the older 'zigzag' model. Hence, I suggest that authors cite a more recent review paper that more adequately describes the extant knowledge in this area.

Response: According to the comment, we have cited a recent review article about R genes-effector interactions and the associated complexity (Kourelis et al. *Plant Cell* **30**, 285-299 (2018)). We have been added a sentence to describe the extent knowledge in discussion section.

Comment 10: Figure 8 should be modified to make clear that the interaction, perhaps better stated as recognition, of NodP by Rj2 could be direct (as currently drawn in the figure) or indirect.

Response: According to the comments, Figure 8 has been modified. We put the words "interaction (with unknown mechanism)" above the arrow connected the symbols of Rj2 and NopP.

Comment 11: I found it interesting that, in the case of 122ΩrhcJ (no T3SS occurs at all), nodule formation and plant growth were almost indistinguishable from USDA 110T. In other words, a variety of other effectors delivered from USDA 122 to Rj2-soybean are not necessary for nodule formation in Rj2-soybean? This is quite a different case than one would find in plant pathogens, for example, and, hence, deserves some discussion in the paper.

Response: As reviewers described, *Bradyrhizobium diazoefficiens* USDA 110 (a model symbiont of soybean bradyrhizobia), have more than 30 candidates of T3SS effectors. Empirically, there are few T3SS effectors in USDA 110 that do not affect nodulation of soybean cultivars as compare with other legumes. An exception is NopE in USDA 110: NopE1 and NopE2 secreted via T3SS have a positive effect on nodulation of soybean to some extents (Wenzel et al. 2010, MPMI 23:124–129). By these backgrounds, we think that nodulation and plant growth of 122ΩrhcJ inoculation were almost indistinguishable from those of USDA 110T.

Comment 12: Regarding Fig. 2.

The authors need to show a picture of the nodule in the root.

Response: The pictures of the nodule have been added to Figure 2b as indicated.

Comment 13: Regarding Fig. 3.

Bradyrhizobium diazoefficiens NopP (244 amino acid) is predicted as approximately 27 kDa.

However, in the Fig 3C, NopP was detected at the 35 kDa. So, NopP is regulated by post-translational gene regulation in Rj2 soybean? This is a particularly problematic part of the paper, especially considering how the data are subsequently discussed (see comment above). A few very easy experiments could be done to help clarify this. For example, does phosphatase treatment shift the band on the gel so that it runs closer to 27 kDa?

Response: As mentioned answer to Comment 6, we thought that NopP secreted into the culture sup were not phosphorylated. Because no band shift by phosphatase treatment was detected by Phos-tag SDS-PAGE technology. *nopP* gene (BD122_09010, 834 bp) in USDA122 was annotated as 277 amino acids length of protein (Genbank accession CP013127). The translational start codon was consistent with *nopP* in the genome of USDA110 that was re-annotated previously (Süß et al. *J Biotechnol* **126**, 69-77 (2006)). Therefore, NopP of both USDA 122 and 110 can be predicted as 31 kDa based on the amino acid sequence. However, we cannot rule out the possibility of post-translational modification other than phosphorylation, since the detected protein (35kDa) is still larger than expected (31 kDa).

Comment 14: 122 Δ nopP showed more nodule number per plant than 122 Ω rhcJ (no T3SS occurs at all). Because of other effectors? Explain why?

Response: There are several reports to show some rhizobial effectors increase nodulation (reviewed by Miwa and Okazaki 2017). In the interaction between soybean and *B. diazoefficiens*, Wenzel et al. (2010, MPMI Vol. 23, pp. 124–129) showed that rhizobial NopE1 and NopE2 secreted via T3SS have a positive effect on nodulation of soybean. Therefore, the differences of nodule number observed between 122 Δ nopP and 122 Ω rhcJ (Fig. 3, Supp Fig. 6) might be caused by other effectors (e. g. NopE1 and NopE2).

Comment 15: Regarding Fig. 4.

The authors should confirm that the expression and secretion (see comment) above of the various NodP forms in either USDA 122 or USDA 110T are equivalent.

Response: According to the reviewer comment, we confirmed NopP secretion from USDA110 and 110nopP122, suggesting that substitutions of NopP on the amino acid residues at positions 60, 67, 173, and 271 does not affect the ability of the bacterium to deliver the protein. We have included this data in Figure 4a, and re-written the sentence in the revised manuscript. Although the transcription levels have not been examined, the amounts of NopP protein detected by Western blotting was almost equal. In addition, 110nopP122 should have completely the same nucleotide sequence of upstream *nopP* (122-type). Therefore, we considered that expression of *nopP* is not different by various NopP forms in USDA 110.

Comment 16: In the figure legend, Scale bars, 1 cm. è Scale bar, 1 cm.

Response: This has been corrected as indicated.

Comment 17: Regarding Fig. 5. The authors should also show GFP photographs in Lee + rj2 and Lee vector control.

Response: The pictures have been added to Figure 5e-l as indicated.

Comment 18: It would be useful to demonstrate downstream gene expression to determine whether a defense response does occur upon treatment with USDA 122 or 122nopP110 in Lee + Rj2 and Lee + rj2, respectively. This might be the most direct way to document that plant immunity is indeed playing a role.

Response: We agree with the reviewer's idea. It is a little difficult experiment and time-consuming. Thus, in the future, we would like to examine a defense response, such as PR genes expression, H₂O₂ accumulation in the roots, occur upon treatment with USDA 122 and 122nopP110 in Lee + Rj2 and Lee + rj2.

Comment 19: Mistakes

Response: All mistakes have been corrected as indicated.

Reviewer #3 (Remarks to the Author):

Comment: In this manuscript by Sugawara and colleagues, the authors identified NopP as an avirulence protein that is necessary for USDA122 to trigger Rj2-dependent incompatibility. Comparisons between NopP alleles suggested that compatibility and incompatibility is due to variation in one three amino acids.

This work is amazing! I am exceptionally enthusiastic about this manuscript because the work is very impactful. It draws parallels between mechanisms of plant-pathogen and plant-mutualist interactions and will definitely influence the thinking in the field. The data suggest that plant immunity may not distinguish beneficial from detrimental symbiont; NopP likely dampens PTI. The data further suggest that plant immunity has key nodes that are susceptible to attack by

microbe-associated effector proteins. I also think it is possible that if Rj2 is a guard, that NopP and a pathogen effector protein likely modify in the same manner, the same guard. That incompatibility and compatibility differ by few amino acids in NopP is novel; in plant-pathogen interactions, it is typically a presence/absence polymorphism in type III effector genes. Last, the results are wholly consistent with the conclusion that type III effector genes of rhizobia exhibit mutualistic co-evolution with host defenses.

The work is simple, yet brilliant. The experiments were logical and comprehensive and the results are convincing. The manuscript is tight and easy to follow. The authors smartly relied on spontaneous mutants and the easily visualized gain of nodulation phenotype to identify causative mutations in NopP. They engineered strains and also relied on natural variation in *nopP* alleles to test their hypothesis. Sugawara and colleagues confirmed the gene-for-gene relationship between *nopP* and Rj2. Last, the authors even cleared up a previous study and showed that NopP has beneficial effects in mung bean.

However, my review is not without some minor comments. I felt the writing needs to be improved in three areas. First, in key places, there needs to be more attention to precision in language. Second, there are just enough errors and awkwardly phrased sentences to be noticeable. Third, the impact could be elevated if the authors were to discuss the broader implications of their work. I am a little worried that the discussion focuses too much on the specific NopP-Rj2 system and does not adequately relate the work to the larger implications of immunity-effector interactions in plant-microbe interactions. The impact of the work may not be sufficiently recognized by members of the wider field. Below are some, but not all, cases:

Response: Thank you for your positive evaluations. According to the first and second comments, we edited precision in English language, errors and awkwardly sentences throughout text as possible as we can. As for the third comments, we obtained additional key results that were required by other reviewers: By incompatible strains, *Rj2*-soybean plant activated a defense gene, PR-2, and suppressed infection thread number, 2 days after inoculation (Fig. 3d, Fig. 6, Suppl. Fig. 8), providing genetic and morphological evidence for defense reaction via ETI. Thus, we changed title including ETI from “Variation of three amino acid residues in bradyrhizobial NopP effector determines symbiotic incompatibility with *Rj2*-soybeans” to “Variation in bradyrhizobial NopP determines symbiotic incompatibility with *Rj2*-soybeans via effector-triggered immunity” for attracting broad readers. In addition, we added and edited new descriptions in the discussion section to emphasize the impact of our work and the implications of immunity-effector interactions in plant-microbe interactions.

Minor Comment 1: Line 128: secretion is not sufficient to call it an effector. It is safer to say that NopP is secreted in a T3SS-dependent manner.

Response: We agree with the reviewer comment. We have been rephrased as indicated.

Minor Comment 2: Line 135: did not test nitrogen fixation (strike these words out).

Response: The data of nitrogen fixation activity have been added to Fig. 3f.

Minor Comment 3: Line 138: change “induces” to “is necessary for”

Response: This has been changed as indicated.

Minor Comment 4: Line 237: this is not clear to me. I am guessing that the authors sequenced the *nopP* alleles of the Is-1 mutants to ask if there were second site mutations in *nopP*?

Response: Yes. We sequence *nopP* in the Tn5-mutants since we suspected there were second mutation sites in the gene.

Minor Comment 5: Errors/awkward:

Line 18

Line 32

Line 43: escape PTI, not the patterns.

Response:

(line 18) the sentence has been corrected to “Genotype-specific incompatibility in legume–rhizobium symbiosis has been suggested to be controlled by effector-triggered immunity underlying pathogenic host-bacteria interactions.”

(Line 32) The sentence has been re-written.

(Line 43) The sentence has been re-written.

Minor Comment 6: I am very supportive of this manuscript. I loved it.

Response: We appreciate your warm support.

Reviewer #1 (Remarks to the Author):

The authors have addressed most of my concerns except for my minor comment #6. A "that-clause" following "suggest" always uses a simple verb form no matter the subject is singular or plural.

Reviewer #2 (Remarks to the Author):

I compliment the authors is doing a nice, thorough job in addressing all of my concerns on their previous submission. I believe the result is a much tighter, more comprehensive paper that should be a significant contribution to the literature.

I have no other critical comments.

Reviewer #3 (Remarks to the Author):

This is a resubmission by Sugawara and colleagues.

I was extremely enthusiastic with the original submission. I remain excited about this work. I wished the authors had devoted a little more attention in the discussion to articulating the impact of the work. Nonetheless, that does not affect my opinion of the article. I still really enjoy it and feel strongly that this work should be published. Below are some minor editorial suggestions.

Page 2; line 28: I would exercise more caution and refer to PR-2 as a marker gene for defense, as opposed to a defense gene.

Page 3; line 49: R proteins do not necessary directly recognize an effector and often recognize the consequence of the action of the effector protein.

Page 4; line 59: make effector plural

Page 4; line 77 (page 5; line 102): cancelled is a usual word choice. How about "overcame" or "evade"?

Page 4; line 78: the sentence essentially says among different amino acids residues, three amino acid residues. I recommend dropping "amino acid residues" from the second part and leave it as "three".

Page 5; line 103, etc. 8 should be written out as eight.

Page 6; line 108: change "responsible for" to "necessary to trigger"

Page 7; line 146: consider changing gene to alleles

Page 8; line 164: bacterium -> bacteria (more than one cell is assayed)

Page 8; line 168: "The number of infection.." is awkward. Please reconstruct.

Page 9; line 188: "Roots transformed with..." -> "Roots transformed with rj2 or empty vector formed nodules regardless of whether it was infected with USDA122 or 122nopP."

Page 12; line 262: "located in THE symbiosis island". I would encourage the authors to remove this figure and discussion from the article. I am not comfortable with the experimental design and the conclusions that were drawn from it. It is a neighbor joining tree. There are unresolved polytomies in the trees. A single gene tree is being compared to concatenated trees: within nopP, there is evidence for many alleles being identical in sequence whereas the variation in the concatenated nod tree may be due to variation of a single gene.

Reviewer #1 (Remarks to the Author):

Comment: The authors have addressed most of my concerns except for my minor comment #6. A "that-clause" following "suggest" always uses a simple verb form no matter the subject is singular or plural.

Response: Thank you very much for your comments to our manuscript. The mistake of grammar has been corrected as indicated.

Reviewer #2 (Remarks to the Author):

Comment: I compliment the authors is doing a nice, thorough job in addressing all of my concerns on their previous submission. I believe the result is a much tighter, more comprehensive paper that should be a significant contribution to the literature. I have no other critical comments.

Response: Thank you very much for your review and comments.

Reviewer #3 (Remarks to the Author):

Comment: This is a resubmission by Sugawara and colleagues.

I was extremely enthusiastic with the original submission. I remain excited about this work. I wished the authors had devoted a little more attention in the discussion to articulating the impact of the work. Nonetheless, that does not affect my opinion of the article. I still really enjoy it and feel strongly that this work should be published. Below are some minor editorial suggestions.

Response: Thank you for your comments. We have edited the descriptions in the Discussion section to articulating the impact of our work according to your previous comments.

Minor comments:

Page 2; line 28: I would exercise more caution and refer to PR-2 as a marker gene for defense, as opposed to a defense gene.

Response: We agree with your suggestion. This has been rephrased "defense gene PR-2" to "defense marker gene PR-2" based on the reviewer comment.

Page 3; line 49: R proteins do not necessary directly recognize an effector and often recognize the consequence of the action of the effector protein.

Response: According to the comment, the sentence has been rephrased (line 41 in the revised manuscript).

Page 4; line 59: make effector plural

Response: This has been corrected based on the reviewer comment (line 59 in the revised manuscript).

Page 4; line 77 (page 5; line 102): cancelled is a usual word choice. How about "overcame" or "evade"?

Response: We agree with the suggestion. These have been changed to "overcome" or "overcoming" in the revised manuscript.

Page 4; line 78: the sentence essentially says among different amino acids residues,

three amino acid residues. I recommend dropping “amino acid residues” from the second part and leave it as “three”.

Response: The sentence has been rephrased as indicated (line 80 in the revised manuscript).

Page 5; line 103, etc. 8 should be written out as eight.

Page 6; line 108: change “responsible for” to “necessary to trigger”

Page 7; line 146: consider changing gene to alleles

Page 8; line 164: bacterium -> bacteria (more than one cell is assayed)

Response: These have been corrected based on the reviewer comment.

Page 8; line 168: “The number of infection..” is awkward. Please reconstruct.

Response: We have changed the sentence to “The number of infection thread (IT) formed on the main roots inoculated with incompatible strains (USDA 122 or 110*nopP*₁₂₂) was significantly less than those with compatible strains (122*nopP*₁₁₀ or USDA 110¹) at 2 days after inoculation (DAI) (Fig. 4e).” (line 170-173 in the revised manuscript)

Page 9; line 188: “Roots transformed with...” -> “Roots transformed with rj2 or empty vector formed nodules regardless of whether it was infected with USDA122 or 122*nopP*.”

Response: The sentence has been rephrased as indicated (line 190-192 in the revised manuscript).

Page 12; line 262: “located in THE symbiosis island”. I would encourage the authors to remove this figure and discussion from the article. I am not comfortable with the experimental design and the conclusions that were drawn from it. It is a neighbor joining tree. There are unresolved polytomies in the trees. A single gene tree is being compared to concatenated trees: within *nopP*, there is evidence for many alleles being identical in sequence whereas the variation in the concatenated nod tree may be due to variation of a single gene.

Response: We agree with the reviewer comment. We have removed the phylogenetic trees based on the concatenated sequences of *nif* and *nod* genes, and discussion associated with phylogenetic relationship of *NopP*.